# Cerebellar modulation of memory encoding in the periaqueductal grey and fear behaviour

**Charlotte Lawrenson\*†, Elena Paci\*†, Jasmine Pickford, Robert AR Drake, Bridget M Lumb, Richard Apps**

School of Physiology, Pharmacology & Neuroscience, University of Bristol, Bristol, United Kingdom

**Abstract:** The pivotal role of the periaqueductal grey (PAG) in fear learning is reinforced by the identification of neurons in male rat ventrolateral PAG (vlPAG) that encode fear memory through signalling the onset and offset of an auditory-conditioned stimulus during presentation of the unreinforced conditioned tone (CS+) during retrieval. Some units only display CS+ onset or offset responses, and the two signals differ in extinction sensitivity, suggesting that they are independent of each other. In addition, understanding cerebellar contributions to survival circuits is advanced by the discovery that (i) reversible inactivation of the medial cerebellar nucleus (MCN) during fear consolidation leads in subsequent retrieval to (a) disruption of the temporal precision of vlPAG offset, but not onset responses to CS+, and (b) an increase in duration of freezing behaviour. And (ii) chemogenetic manipulation of the MCN-vlPAG projection during fear acquisition (a) reduces the occurrence of fear-related ultrasonic vocalisations, and (b) during subsequent retrieval, slows the extinction rate of fear-related freezing. These findings show that the cerebellum is part of the survival network that regulates fear memory processes at multiple timescales and in multiple ways, raising the possibility that dysfunctional interactions in the cerebellar-survival network may underlie fear-related disorders and comorbidities.

**\*For correspondence:**
pycll@bristol.ac.uk (CL);
elena.paci@bristol.ac.uk (EP)

†These authors contributed equally to this work

**Competing interest:** The authors declare that no competing interests exist.

## Editor's evaluation

This study describes interactions between the cerebellum and periaqueductal grey during fear conditioning behavior in rats. The authors have used a combination of electrophysiology, behavioral paradigms, and DREADDs to uncover critical circuits that expand the role of the cerebellum beyond motor function. The results have far reaching implications as they add new context for how inter-regional connections drive complex behaviors and they will likely stimulate new ideas on important brain circuits that cause defects in neurological and neuropsychiatric diseases.

## Introduction

The periaqueductal grey (PAG) lies at the core of central networks that coordinate survival, including coping behaviours mediating defensive and fear-evoked responses. Neurons in the functionally distinct longitudinal columns of the PAG coordinate different aspects of survival behaviours. Of particular relevance to the current study, fear-evoked freezing is mediated by neurons in its ventrolateral sector (vlPAG; *Vianna et al., 2001*; *Walker and Carrive, 2003*; *Tovote et al., 2016*; *Watson et al., 2016*). Fear-related behaviours coordinated by the PAG also extend to the expression of 22 kHz ultrasonic vocalisations (USVs; *Kim et al., 2013*; *Ouda et al., 2016*) and risk assessment activity such as rearing (*Sandner et al., 1987*; *Clelland et al., 2009*). Furthermore, at a cellular level, electrophysiological

**eLife digest** Anxiety disorders are a cluster of mental health conditions characterised by persistent and excessive amounts of fear and worry. They affect millions of people worldwide, but treatments can sometimes be ineffective and have unwanted side effects. Understanding which brain regions are involved in fear and anxiety-related behaviours, and how those areas are connected, is the first step towards designing more effective treatments.

A region known as the periaqueductal grey (or PAG) sits at the centre of the brain's fear and anxiety network, regulating pain, encoding fear memories and responding to threats and stressors. It also controls survival behaviours such as the 'freeze' response, when an animal is frightened.

A more recent addition to the fear and anxiety network is the cerebellum, which sits at the base of the brain. Two-way connections between this region and the PAG have been well described, but how the cerebellum might influence fear and anxiety-related behaviours remains unclear.

To explore this role, Lawrenson, Paci et al. investigated whether the cerebellum modulates brain activity within the PAG and if so, how this relates to fear behaviours. Rats had electrodes implanted in their brains to record the activity of nerve cells within the PAG. A common fear-conditioning task was then used to elicit 'freeze' responses: a sound was paired with mild foot shocks until the animals learned to fear the auditory signal. In the rats, a subset of neurons within the PAG responded to the tone, consistent with those cells encoding a fear memory. But when a drug blocked the cerebellum's output during fear conditioning, the timing of the PAG response was less precise and the rats' freeze response lasted longer.

Lawrenson, Paci et al. concluded that the cerebellum, through its interactions with the brain's fear and anxiety network, might be responsible for coordinating the most appropriate behavioural response to fear, and how long 'freezing' lasts.

In summary, these findings show that the cerebellum is a part of the brain's survival network which regulates fear-memory processes. It raises the possibility that disruption of the cerebellum might underlie anxiety and other fear-related disorders, thereby providing a new target for future therapies.

studies have found that neurons in vlPAG encode associatively conditioned fear memory (*Watson et al., 2016*; *Wright et al., 2019*).

Central nervous system survival networks involving the PAG are well documented (*Carrive et al., 1997*; *Vianna et al., 2001*; *Walker et al., 2002*; *Tovote et al., 2015*; *Watson et al., 2016*; *Ozawa et al., 2016*). Until recently, the cerebellum was generally considered not to be a part of this network. However, there is good evidence in cats and rodents that vermal regions of the cerebellum contribute to motor and autonomic components of defensive states: inactivation of vermal cerebellar cortex (lobules IV–VIII), or one of its main output nuclei (medial cerebellar nucleus [MCN], aka fastigial nucleus), leads to deficits in fear-related behaviours such as context-conditioned bradycardia (*Supple and Leaton, 1990*) the expression of innate fear (*Supple et al., 1987*; *Koutsikou et al., 2014*); and fear-conditioned freezing behaviour (*Asdourian and Frerichs, 1970*; *Sacchetti et al., 2002*; *Sacchetti et al., 2005*; *Koutsikou et al., 2014*). In addition, the emission of USVs has been related to cerebellar function (*Fujita et al., 2008*; *Fujita-Jimbo and Momoi, 2014*; *Toledo et al., 2019*), and inactivation of the cerebellar vermis during innate and conditioned fear unmasks risk assessment rearing behaviour (*Koutsikou et al., 2014*).

The cerebellum is reciprocally connected to many brain regions associated with survival networks (*Sacchetti et al., 2009*; *Strick et al., 2009*; *Apps and Strata, 2015*), and in all mammalian species studied so far (cat, rabbit, rat, mouse, human), this includes interconnections with the vlPAG (*Whiteside and Snider, 1953*; *Teune et al., 2000*; *Nisimaru et al., 2013*; *Koutsikou et al., 2014*; *Cacciola et al., 2019*; *Vaaga et al., 2020*; *Frontera et al., 2020*). Thus, the dependence of survival behaviours on the integrity of the cerebellum, together with reciprocal connections between the vlPAG and the cerebellum, raises the possibility of an important role of cerebellar interactions with the PAG in the expression of such behaviours.

An emerging concept is that the cerebellar vermis and its output nucleus MCN are involved in the control of an integrated array of fear-related functions, including fear memory and fear-induced behaviours such as freezing and their extinction (*Sacchetti et al., 2002*; *Sacchetti et al., 2005*;

*Koutsikou et al., 2014*; *Utz et al., 2015*). Important insights into the role of the murine MCN-vlPAG pathway in modulating PAG fear learning and memory have been provided recently (*Vaaga et al., 2020*; *Frontera et al., 2020*). However, it is not known whether neural encoding of fear memory within PAG is dependent on the integrity of its cerebellar input. More generally, given the well-established role of the cerebellum in the coordination of movements and, in particular, the representation of temporal relationships (e.g. *Ivry, 1997*; *Xu et al., 2006*; *Koch et al., 2006*; *Bares et al., 2007*; *Spencer and Ivry, 2013*; ; *Johansson et al., 2016*), it remains to be determined whether the cerebellum enables the survival circuit network to elicit behaviourally suitable responses at appropriate times during fear conditioning. Effects during retrieval and subsequent extinction of a fear-conditioned response are of particular interest because deficits in extinction processes are thought to underlie psychological conditions such as post-traumatic stress disorder (PTSD) (*Bremner et al., 1999*; *Herry et al., 2010*; *Milad and Quirk, 2012*) and have also been related to chronic pain phenotype (*Ji et al., 2018*).

The experiments reported here used a combination of electrophysiological, behavioural, and interventionist approaches in an auditory cued fear conditioning paradigm in rats to interrogate the role of MCN interactions with the PAG in the expression of fear-related behaviours. Our main findings are that (i) PAG encodes temporally precise information about the onset and offset of a fear-conditioned auditory stimulus and that these two neural signals may be generated by independent mechanisms; (ii) both onset and offset responses correlate with fear-related freezing behaviour; (iii) MCN is part of a larger circuit that regulates the temporal accuracy of PAG encoding of fear-conditioned stimulus offset but not onset during retrieval of a conditioned response; and (iv) MCN is involved in distinct aspects of survival behaviour at different times during fear conditioning: during acquisition, the emission of USVs and subsequent rate of extinction of conditioned freezing during retrieval. While during consolidation its influence on survival circuits affects the duration of conditioned freezing in subsequent retrieval. In summary, the present study provides evidence that the cerebellum through its interactions with survival circuits regulates the ability of the PAG to encode a fear memory trace with temporal precision and also regulates the timing of appropriate patterns of behaviour during fear acquisition and retrieval.

## Results

### vlPAG neuronal responses during an auditory cued fear conditioning paradigm

As a first step, an auditory cued fear conditioning paradigm (*Figure 1A*) was used to investigate how single-unit activity in the PAG responded to fear acquisition and subsequent retrieval and extinction of fear-conditioned responses (n = 10 animals total; n = 6 with dual microdrives and n = 4 microdrives combined with cannulae, see Materials and methods for details and *Figure 1B–E* and *Figure 1— figure supplement 1A–C*). In all animals, the position of tetrodes was histologically verified. The majority of PAG tetrode recording sites were located in the ventrolateral sector of the PAG (vlPAG, so this term will be used, *Figure 1—figure supplement 2*). Note, however, that we cannot rule out inclusion of other ventral recording sites.

A total of 32 vlPAG units (obtained from eight animals) were recorded during habituation (*Figure 1B*). The majority (75%, n = 24/32 units) were unresponsive to the unconditioned auditory tone; the remainder responded (see Materials and methods for definition) to tone onset (either increasing, n = 6/32 units, or decreasing firing rate, n = 2/32 units); and in some of these cases (15.6%) also to tone offset (either increasing 3/32 units or decreasing firing rate 2/32 units).

We recorded the activity of 50 vlPAG units (obtained from 10 animals) during acquisition where the CS tone was paired with the unconditioned stimulus (US footshock at tone offset (*Figure 1C*)). Some of these units (18.5 %) responded to CS onset. Following the US, 20% of the total also displayed an increase in firing rate which presumably reflects sensory afferent drive to the PAG as a result of the aversive peripheral stimulus (e.g. *Sanders et al., 1980*; *Heinricher et al., 1987*; *Sharma et al., 1999*). Across all recorded units, the average increase in activity following the US was not statistically significant (*Figure 1C*, z-score < 1.96).

During retrieval and extinction of the conditioned response, we recorded a total of 55 vlPAG units (obtained from 10 animals). Retrieval is defined here as early extinction (EE) training when the unreinforced CS+ reliably elicits a conditioned response. In retrieval, 29 units (52.7%, from n = 9 animals)

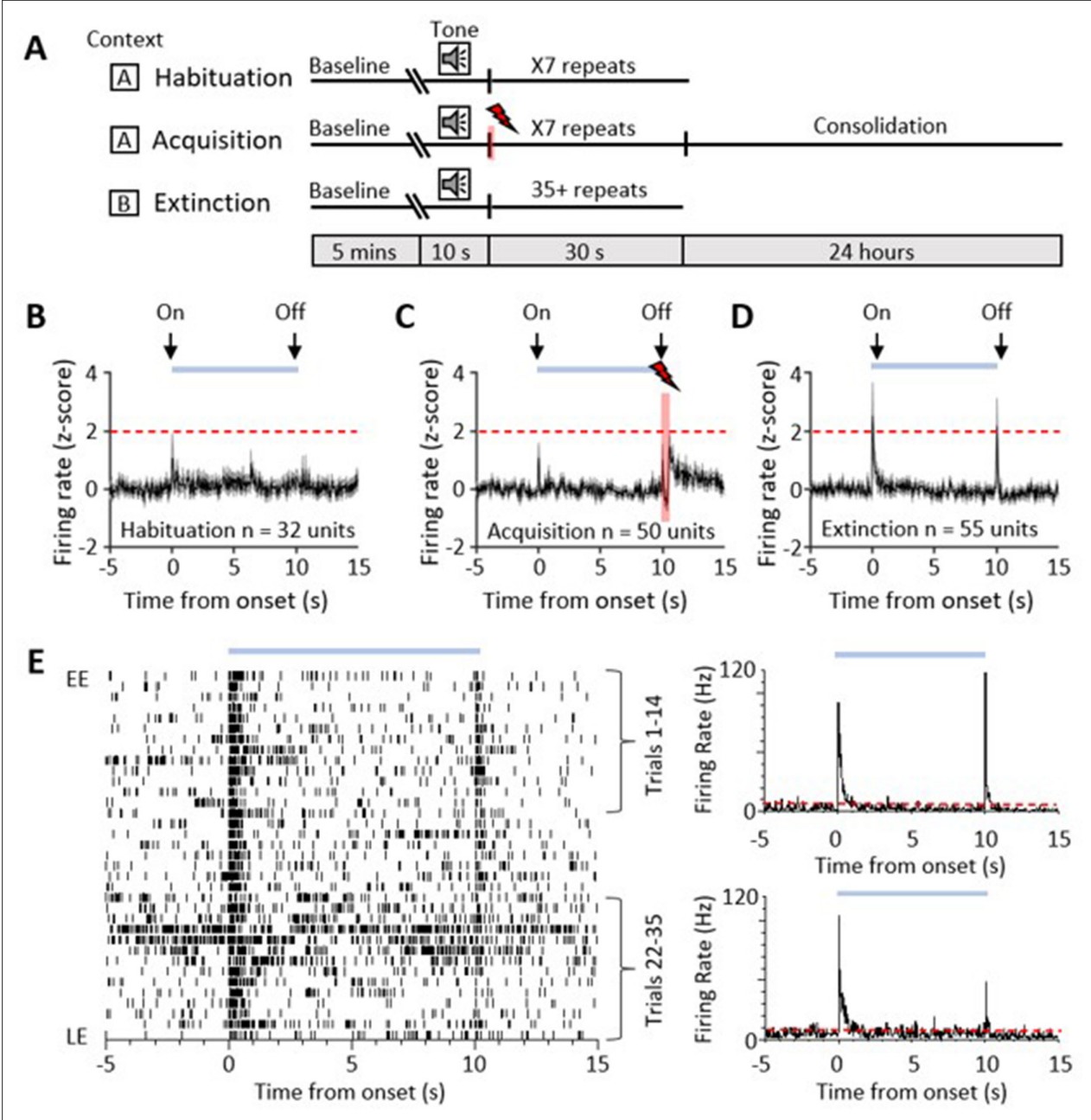

**Figure 1.** Single-unit ventrolateral periaqueductal grey (vlPAG) responses during auditory cued fear delay conditioning. (**A**) Schematic representation of the fear conditioning paradigm composed of habituation, acquisition, and extinction sessions. Habituation and acquisition were carried out in context A, whilst extinction training was in context B. During acquisition, a conditioned stimulus (CS) tone was paired with an unconditioned stimulus (US) footshock (see Materials and methods for details). (**B**) Peri event time histogram (PETH) showing average firing rate of all available single units (n = 32) recorded during presentation of the unconditioned tone (10 s) during habituation in eight control animals. Time 0, tone onset. Light blue bar shows time the tone is on. (**C**) Same as (**B**) but all available single units (n = 50) recorded during acquisition from 10 control animals (shaded red and lightening symbol indicates time of US footshock when the stimulus artefact prevented neural recording). (**D**) Same as (**B**) but all available single units (n = 55) recorded during presentation of the unreinforced CS+ throughout extinction from 10 control animals. For (**B**–**D**), individual unit activity was z-score normalised to a 5 s baseline before tone onset. Horizontal dashed red line represents significance level (p<0.05). PETHs show mean ± SEM; 40 ms bins. (**E**) One example of a type 1 onset and offset single unit recorded during extinction training. Data displayed as raster plot from early extinction (EE) to late extinction (LE) with corresponding PETH for EE (trials 1–14) and LE (trials 22–35) (40 ms bins), time 0 onset of CS+. Horizontal dashed red line represents significance level (p<0.05).

The online version of this article includes the following source data and figure supplement(s) for figure 1:

**Figure supplement 1.** Tetrode recordings.

*Figure 1 continued on next page*

*Figure 1 continued*

**Figure supplement 1—source data 1.** Numerical data to support graphs in *Figure 1—figure supplement 1*.

**Figure supplement 2.** Histological verification of implants.

**Figure supplement 3.** Ventrolateral periaqueductal grey (vlPAG) offset responses during auditory cued fear trace conditioning.

**Figure supplement 3—source data 1.** Numerical data to support graphs in *Figure 1—figure supplement 3*.

displayed a transient increase in activity during presentation of the unreinforced conditioned stimulus (CS+, *Figure 1D and E*). In keeping with a previous classification of PAG unit activity during fear-conditioned extinction training (*Watson et al., 2016*), these units are defined as type 1. The pattern of response was typically a phasic increase in activity at CS+ onset, but an additional feature not previously reported for vlPAG units was a phasic increase in activity at CS+ offset (*Figure 1D and E*). A total of 21/29 type 1 units (72.4%) responded to both CS+ onset and CS+ offset, while 3/29 units (10.3%) only responded to CS+ onset and 5/29 units only responded to CS+ offset (17.3%). This raises the possibility that CS+ onset and offset responses may be mediated by separate mechanisms. We have therefore termed them type 1 onset and type 1 offset responses, respectively.

Of the remainder of the sample of vlPAG units recorded in control animals during EE, 22 units (40%) demonstrated no significant change in firing rate and were therefore defined as type 2 (*Watson et al., 2016*). Four units (7%) responded to CS+ onset with a decrease in firing rate (two of these units also reduced their firing rate at CS+ offset) and therefore were classed as type 4. No type 3 units (biphasic response) were observed.

Since the majority of responsive vlPAG units recorded in control experiments were type 1 (i.e. displayed an increase in firing rate), the following analysis is confined mainly to consideration of their activity during fear-conditioned retrieval and extinction. The majority of type 1 onset units (75%, 18/24 units) and type 1 offset units (73%, 19/26 units) showed a significant reduction in responsiveness during extinction training, as measured by the integrated area of response during CS+ trials in EE versus late extinction (LE, *Figure 2B and D*, control, type 1 onset units: $t(17) = 5.146$, $p<0.0001$, n = 18, paired *t*-test; type 1 offset units, *Figure 2F and H*: $t(18) = 4.4$, $p=0.0003$, n = 19, paired *t*-test). For type 1 onset responses, this reduction was not evident for mean peak z-score (*Figure 2E*, decrease of 13.8% from EE to LE; $t(17) = 1.344$, $p=0.197$, n = 18, paired *t*-test) but was evident for type 1 offset responses (*Figure 2I*, decrease of 67.0% from EE to LE; $t(18) = 2.354$, $p=0.0302$, n = 19, paired *t*-test). This distinction between onset and offset responses provides additional evidence that they may be evoked by separate mechanisms.

## vlPAG offset responses during trace conditioning

For the results described above, a delay classical conditioning paradigm was used where the US was timed to occur at CS offset (*Figure 1A*, see Materials and methods). For the unit activity recorded at CS+ offset, it was therefore unclear whether the change in firing rate during retrieval signals the end of the conditioned tone and/or represents the timing of the expected US. To examine this, in three animals, a 1 s trace interval was introduced between the CS and US during acquisition trials. A total of six type 1 units (those with increased activity at CS+ offset) were found to show a change in activity that was temporally related to the offset of the CS+ during retrieval and not to the time of the expected occurrence of the US (1 s after CS+ offset, *Figure 1—figure supplement 3A and B*). We also recorded six units with type 4 responses (those with reduced activity at CS+ offset) and in trace conditioning their activity was also related to time of CS+ offset (*Figure 1—figure supplement 3C and D*). A comparison of time to peak or trough of response for all available units recorded during control versus trace conditioning showed no statistical difference (*Figure 1—figure supplement 3E*; $t(25) = 1.76$, $p=0.091$, unpaired *t*-test). These results are therefore consistent with vlPAG activity signalling the conditioned tone offset rather than a prediction of the timing of the US.

## Population activity in the MCN and vlPAG

It was also of interest to consider any changes in neural population activity during extinction training. To assess such changes, auditory event-related potentials (ERPs) were recorded simultaneously from the MCN and vlPAG, following a delay conditioning paradigm (n = 6 animals, *Figure 3*). For vlPAG, we

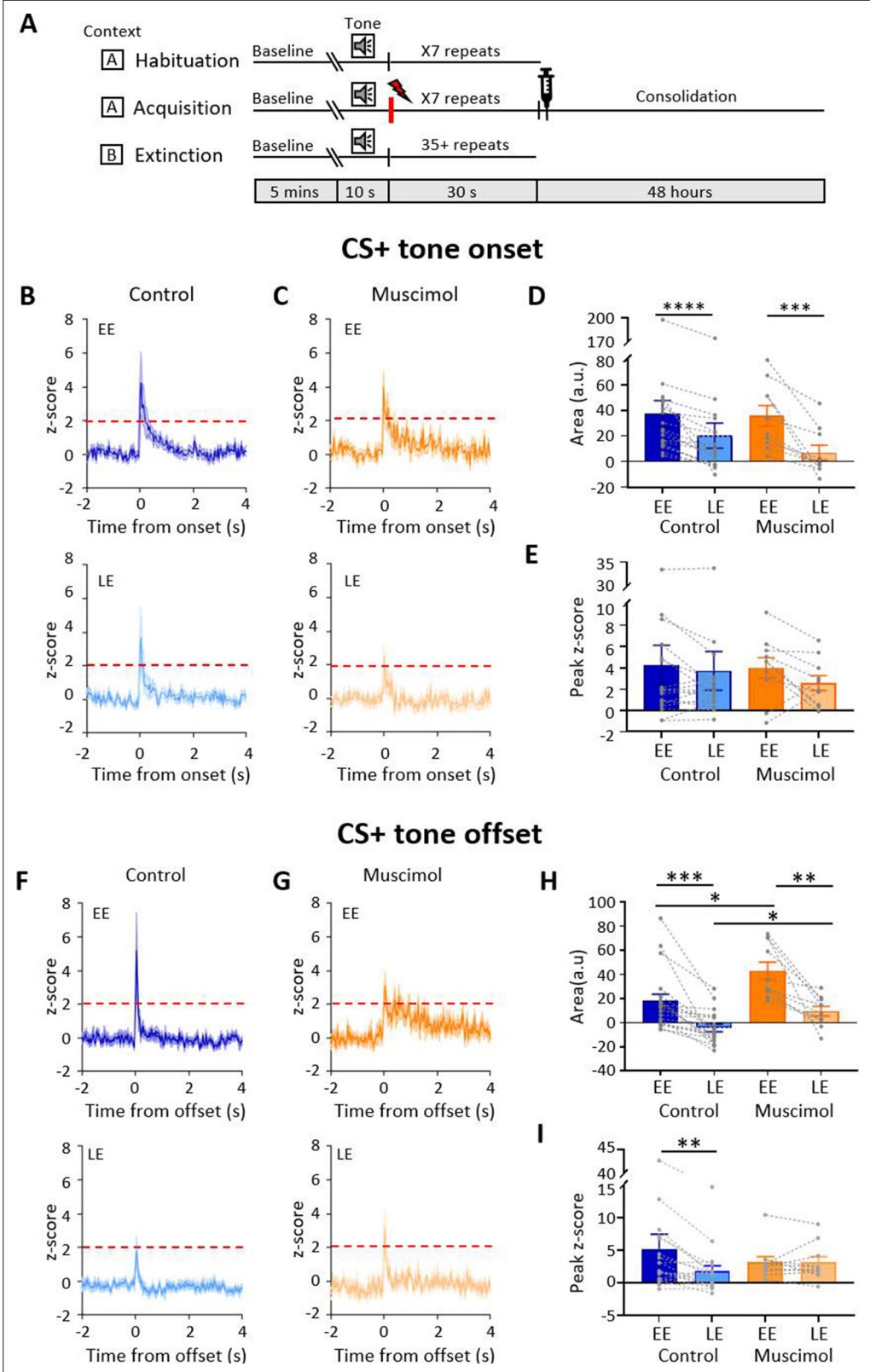

**Figure 2.** Effect of medial cerebellar nucleus (MCN) inactivation during consolidation on ventrolateral periaqueductal grey (vlPAG) type 1 onset and offset responses during extinction. (**A**) Schematic timeline representing the fear conditioning protocol for muscimol animals. (**B**) Group data for control animals showing average z-scored type 1 onset responses (n = 18 single units from n = 7 animals) during early extinction (EE, upper

*Figure 2 continued on next page*

*Figure 2 continued*

panel) and late extinction (LE, lower panel); solid lines in each plot show mean z-score, shaded regions ± SEM; horizontal dashed red lines show significance level (p<0.05). Time 0, CS+ tone onset. (**C**) Same as (**B**) but grouped data for muscimol animals (n = 10 single units from n = 3 animals). (**D**) Bar charts showing average type 1 onset response area (arbitrary units) during EE and LE for single units recorded in control versus muscimol animals. Pairs of data points connected with dashed lines show change in mean response area for each single unit over extinction training. Bars show group means ± SEM. Paired *t*-test, ****p<0.0001, ***p<0.001. (**E**) Same as (**D**) but grouped data for peak z-score. (**F**) Same as (**B**) but control type 1 offset responses (n = 19 single units from n = 7 animals) during EE (upper panel) and LE (lower panel). Time 0, CS+ tone offset. (**G**) Same as (**F**) but grouped data for muscimol animals (n = 10 units from n = 3 animals). (**H**) Same as (**D**) but showing average type 1 offset response area (arbitrary units) during EE and LE for single units recorded in control versus muscimol animals. Paired *t*-test, ***p<0.001, **p<0.01; unpaired *t*-test, **p<0.01, *p<0.05. (**H**) Same as (**I**) but grouped data for peak z-score. Paired *t*-test, *p<0.05.

The online version of this article includes the following source data and figure supplement(s) for figure 2:

**Source data 1.** Numerical data to support graphs in *Figure 2*.

**Figure supplement 1.** Comparison of single-unit results for tetrodes-only versus saline control animals.

**Figure supplement 1—source data 1.** Numerical data to support graphs in *Figure 2—figure supplement 1*.

were able to compare the ERP waveform to the simultaneously recorded unit activity and found that field duration and spike activity were broadly concomitant (*Figure 3—figure supplement 1*).

The ERP recorded in the vlPAG at CS+ onset had a significantly shorter onset latency than the ERP recorded in the same animals in MCN (vlPAG onset 6.5 ± 1.45 ms; MCN onset 25.2 ± 4.05 ms, t(10) = 4.33, p=0.007, paired *t*-test). However, latency to peak was not significantly different (PAG peak 58.5 ± 7.4 ms; MCN peak 84.7 ± 11.46 ms, t(5) = 1.418, p=0.215, paired *t*-test). At CS+ offset, the ERP in the vlPAG was also significantly shorter in latency than the ERP in MCN (vlPAG offset 32 ± 5.2 ms; MCN offset 47.7 ± 3.3 ms, t(5) = 2.991, p=0.0304; paired *t*-test), while latency to peak was similar (vlPAG peak 99 ± 6.4 ms; MCN peak 100.7 ± 6.14 ms, t(5) = 1.746, p=0.141, paired *t*-test). The difference in onset latency between ERP responses recorded at CS+ onset and offset within both vlPAG and MCN was statistically significant (vlPAG onset vs. offset, t(10) = 4.720, p=0.0008, unpaired *t*-test; MCN onset vs. offset, t(10) = 4.319, p=0.0015, unpaired *t*-test). These findings indicate that both MCN and vlPAG receive auditory inputs that convey information about conditioned tone onset and offset. Given the likely differences in central pathways involved, the disparity in latency of the ERPs recorded within the two brain regions is perhaps unsurprising (*Huang et al., 1982*; *Vianna and Brandão, 2003*; *Wang et al., 2019*). However, the difference in latency between ERP responses recorded at CS+ onset compared to CS+ offset within each structure suggests that they are also likely to be generated by different neural pathways.

During extinction training, the peak-to-peak amplitude of the ERPs recorded in the MCN and vlPAG at CS+ onset did not show a statistically significant difference (*Figure 3A and C*; MCN: EE ERP 421.6 ± 126 mV vs. LE ERP 309.3 ± 88.7 mV, t(5) = 1.905, p=0.115, paired *t*-test; vlPAG: EE ERP 501.1.2 ± 104.7 mV vs. LE ERP 418 ± 84.08 mV, t(5) = 1.122, p=0.313, paired *t*-test). By comparison, ERPs recorded at CS+ offset in the vlPAG (*Figure 3B and D*) showed a significant decrease in amplitude during extinction training in EE versus LE (65% reduction in ERP size, EE 520.3 ± 124.7 vs. LE 337.8 ± 96.29, t(5) = 2.258, p=0.007, paired *t*-test), while in the MCN there was no significant decrease (EE 537.3 ± 173.2 vs. LE 430.5 ± 135.6, t(5) = 2.258, p=0.074, paired *t*-test). The differences found in vlPAG between onset and offset ERPs provide further evidence that they are generated and regulated by different neural pathways.

To assess if changes in ERP amplitude during extinction covaried between MCN and vlPAG, we compared average ERP amplitude at CS+ onset (*Figure 3E*) and CS+ offset (*Figure 3F*) for the five blocks of extinction training. There was a significant correlation between the amplitude of ERP responses recorded in MCN and vlPAG for all available cases (n = 6 animals) for both CS+ onset and CS+ offset (onset, $r_{rm}$ = 0.64, p<0.0001; offset, $r_{rm}$ = 0.44, p=0.033; repeated-measures correlation).

Taken together, these ERP data therefore suggest that population activity in MCN and vlPAG broadly parallels the changes that occur in vlPAG single-unit peak activity at onset and offset of the CS+ during extinction of a fear-conditioned response.

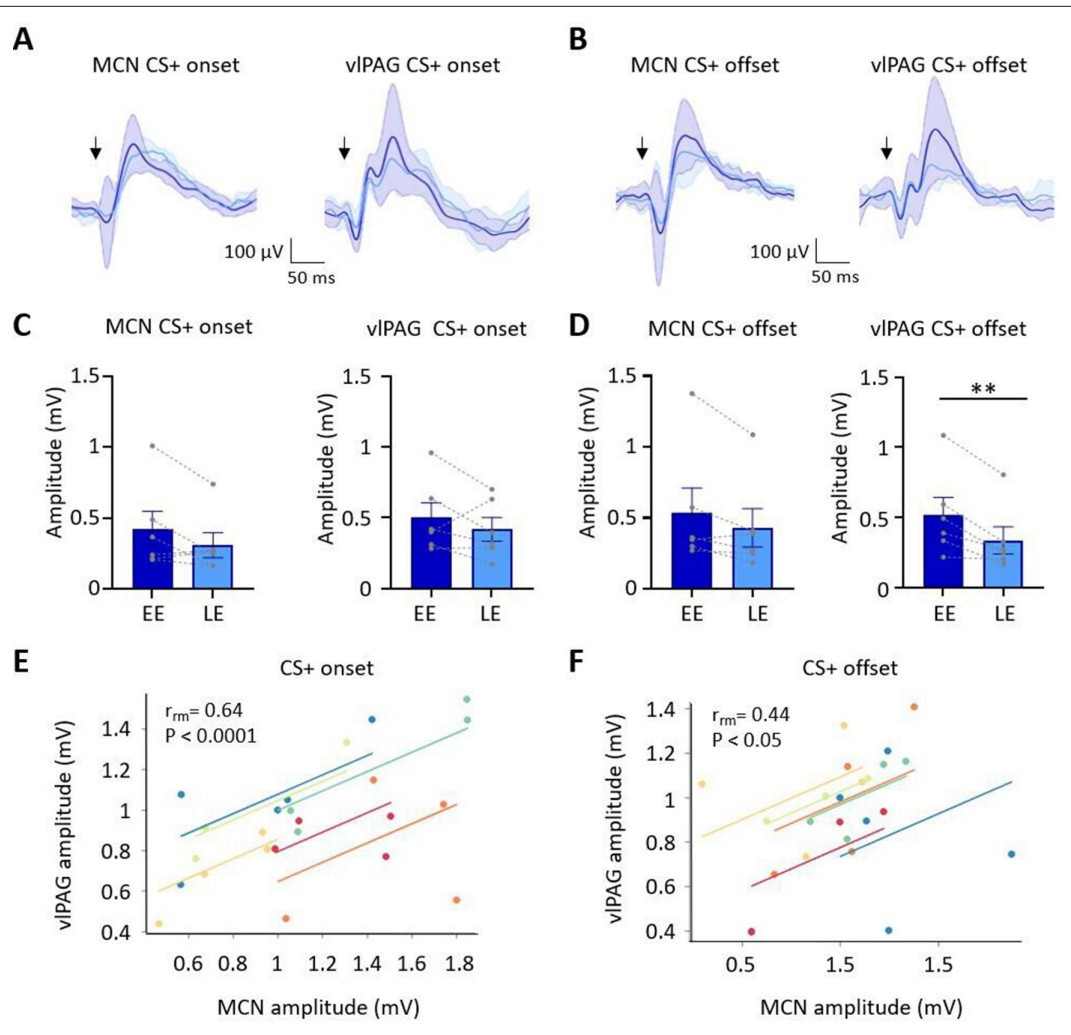

**Figure 3.** Auditory event-related field potentials (ERPs) recorded simultaneously in the medial cerebellar nucleus (MCN) and ventrolateral periaqueductal grey (vlPAG) during extinction. (**A**) Group average ERPs recorded at CS+ onset in the MCN and vlPAG in control animals (n = 6 rats), arrows indicate time of tone onset; each waveform shows mean ± SEM; dark blue, average ERP during early extinction (EE), light blue, average ERP during late extinction (LE). (**B**) Same as (**A**) but ERPs recorded simultaneously at CS+ offset (n = 6 rats). (**C**) Plots showing mean peak-to-peak amplitude of ERPs recorded at CS+ onset in EE versus LE; left panel, MCN; right panel, vlPAG (n = 6 rats, means ± SEM). Pairs of data points connected with dashed lines show the change in mean amplitude over extinction training for individual animals. (**D**) Same as (**C**) but for CS+ offset (n = 6 rats, paired *t*-test, **p<0.01). (**E**) Repeated-measures correlation ($r_{rm}$) for ERPs at CS+ onset (n = 6 rats) comparing the amplitude of ERPs recorded simultaneously in MCN and vlPAG. Each colour represents data and line of best fit for an individual animal. (**F**) Same as (**E**) but for CS+ offset (n = 6 rats).

The online version of this article includes the following source data and figure supplement(s) for figure 3:

**Source data 1.** Numerical data to support graphs in *Figure 3*.

**Figure supplement 1.** Relationship between unit activity and event-related potentials (ERPs).

## The effect of temporary MCN inactivation on vlPAG activity during fear consolidation

In eight additional animals, a delay conditioning paradigm was used (see *Figure 2A*, n = 4 rats with bilateral cannulae in MCN and unilateral tetrodes in PAG, and n = 4 rats with bilateral cannulae only in MCN; see *Figure 1—figure supplement 2F and G*), and muscimol was infused into the MCN to reversibly block cerebellar output during consolidation of the fear-associative memory prior to extinction training (termed 'muscimol' in extinction sessions, *Figures 2, 4 and 5*; see also *Figure 1—figure*

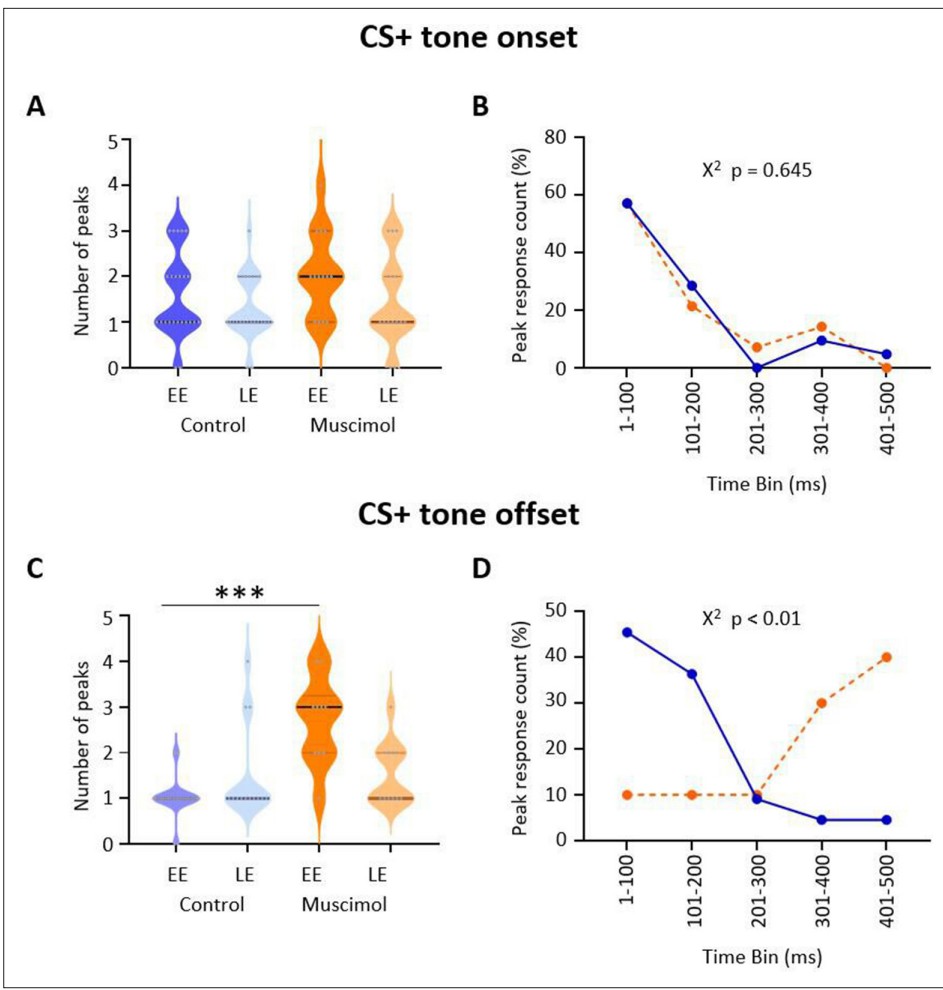

**Figure 4.** Effect of medial cerebellar nucleus (MCN) inactivation during consolidation on the timing of ventrolateral periaqueductal grey (vlPAG) type 1 onset and offset responses during extinction. (**A**) Violin plots showing the number of significant peaks of activity (≥2 SD from baseline) that occur in the initial 500 ms following CS+ onset for early extinction (EE) and late extinction (LE) in control rats (blue, n = 18 units) and muscimol rats (orange, n = 10 units). (**B**) Proportion of peak responses as a function of time after CS+ onset. For each unit, the 100 ms time bin in which the maximum peak response occurred in the first 500 ms after CS+ onset was expressed as a percentage of total count during EE. Control (blue); muscimol (orange). Chi-squared test, p>0.05. (**C**) Same as (**A**) but for CS+ offset. ***p<0.001, unpaired *t*-test. Control (blue, n = 19); muscimol (orange, n = 10). (**D**) Same as (**B**) but for CS+ offset. Chi-squared test, p<0.01.

The online version of this article includes the following source data for figure 4:

**Source data 1.** Numerical data to support graphs in *Figure 4*.

---

*supplement 2F and G*). After a delay of 48 hr, to ensure complete washout of the drug, the animals were exposed to the unreinforced CS+ in extinction training. There was no significant difference between the overall firing rates across habituation, acquisition, and extinction training (*Figure 1— figure supplement 1D*, F(2, 49) = 0.506, p=0.079, n = 52, one-way ANOVA).

In the four muscimol animals with tetrodes, a total of 14 type 1 vlPAG units were recorded from three animals during extinction. Units were recorded from the fourth animal but were not type 1 responses so were not analysed further. Similar to the control results described above, the majority (71%, 10/14 units) of available type 1 onset units (*Figure 2C*) showed a reduction in responsiveness during extinction training as measured by mean change in integrated area of response (*Figure 2D*; on average, the integrated area of response was 77.8% smaller between EE and LE, t(9) = 4.294, p=0.002, n = 10, paired *t*-test). In keeping with the findings from control animals, no significant difference was found for mean peak z-score (*Figure 2E*, peak 0.7% smaller between EE and LE, t(9) = 1.707,

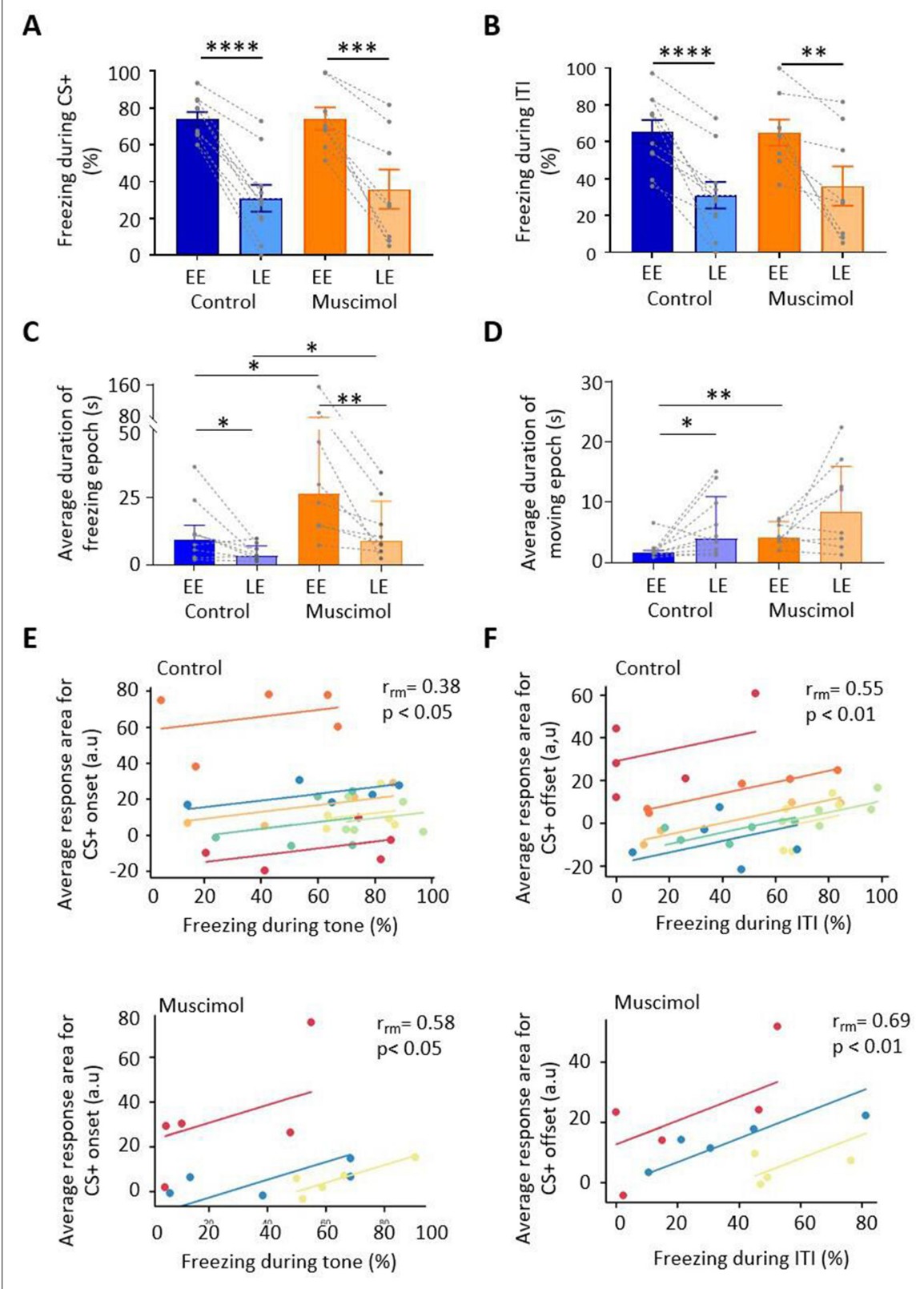

**Figure 5.** Freezing activity during extinction for control and muscimol animals. (**A**) The percentage of total time the conditioned tone (CS+) was presented during early extinction (EE) and late extinction (LE) that freezing epochs occurred in control (n = 10 rats) versus muscimol (n = 8 rats). Pairs of data points connected with dashed lines show the change in mean percentage freezing per animal over extinction training. Bars show group median ± IQR. ****p<0.0001; ***p<0.001. (**B**) Same as (**A**) but for inter-tone interval (ITI). (**C**) Same as (**A**) but for the average duration of time that animals

*Figure 5 continued on next page*

*Figure 5 continued*

displayed freezing behaviour after the CS+. Bars show group means ± SEM. Paired *t*-test. **p<0.01, *p<0.05. (**D**) Same as (**C**) but for average duration of moving epochs. (**E**) Repeated-measures correlation (r$_{rm}$) for control (upper panel, n = 7 rats) and muscimol (lower panel, n = 3 rats), comparing the integrated response area of units (arbitrary units, a.u.) at CS+ onset as a function of percentage of time freezing during presentation of the CS+. Each colour represents data and line of best fit for an individual animal. (**F**) Same as (**E**) but unit activity at CS+ offset in relation to time freezing during ITI.

The online version of this article includes the following source data and figure supplement(s) for figure 5:

**Source data 1.** Numerical data to support graphs in *Figure 5*.

**Figure supplement 1.** Trial-by-trial comparison of conditioned behaviour in muscimol and control animals.

**Figure supplement 1—source data 1.** Numerical data to support graphs in *Figure 5—figure supplement 1*.

**Figure supplement 2.** Conditioned fear-related behaviours.

**Figure supplement 2—source data 1.** Numerical data to support graphs in *Figure 5—figure supplement 2*.

p=0.122, n = 10, paired *t*-test). There was also no statistically significant difference between control and muscimol animals for type 1 CS+ onset responses in EE or LE for either measure of response (*Figure 2D and E*, for control n = 18 vs. muscimol n = 10; mean integrated area in EE, t(26) = 0.112, p=0.912, unpaired *t*-test; mean integrated area in LE, t(26) = 0.949, p=0.351, unpaired *t*-test; mean peak z-score in EE, t(26) = 0.103, p=0.919, unpaired *t*-test; mean peak z-score in LE, t(26) = 0.454, p=0.654, unpaired *t*-test), nor was there a difference in overall firing rates for all units during extinction training (control vs. muscimol; t(78) = 0.407, p=0.685, unpaired *t*-test; *Figure 1—figure supplement 1C and D*).

By comparison to controls (*Figure 2F*, n = 23 units), inspection of *Figure 2G* (n = 10 units) shows that there was, however, a significant difference in muscimol experiments during EE in the average pattern of response of vlPAG units at CS+ offset. The mean integrated area of response in muscimol experiments increased by 58.5% by comparison to control (*Figure 2H*, for control n = 19 vs. muscimol n = 10, t(27) = 2.645, p=0.014, unpaired *t*-test). There was also a reduction in mean peak z-score at EE (*Figure 2G*, muscimol mean peak was 60% of control), but this was not statistically significant (*Figure 2I*, t(27) = 0.660, p=0.515, unpaired *t*-test). In LE, the mean integrated area of response was also significantly different. On average, muscimol experiments had a significantly increased area of response than control animals (41.6% larger, *Figure 2H*, t(27) = 2.365, p=0.014, unpaired *t*-test). Similar to CS+ onset, the mean peak z-score in LE was not significantly different between muscimol and control experiments (*Figure 2I*, t(27) = 1.013, p=0.32, unpaired *t*-test).

To investigate further the difference in pattern of neural response between muscimol and control animals, we plotted for each unit the number of significant peaks of activity in the first 500 ms following CS+ onset and CS+ offset (see Materials and methods for details). After CS+ onset, there was no significant difference in total number of peaks between muscimol and control animals (*Figure 4A*; EE, t(36) = 1.570, p=0.125; LE, t(37) = 0.651, p=0.519; unpaired *t*-test) nor latency to peak (*Figure 4B*; $\chi^2$(4) = 2.50, p=0.645, chi-square), with both groups having about 85% of peaks occurring in the 200 ms immediately following CS+ onset.

By contrast, following CS+ offset both the number (*Figure 4C*; t(31) = 7.184, p<0.0001 unpaired *t*-test) and latency (*Figure 4D*; $\chi^2$(4) = 13.31, p<0.01, chi-square) of peaks significantly increased during EE in muscimol animals. In LE, there was no increase in the number of peaks (t(22) = 0.019, p=0.98). In control animals, 82% of peaks occurred in the 200 ms immediately following the CS+ offset, while only 20% of muscimol animals had peaks during the same time period.

Taken all together, these data therefore suggest that during extinction training the pattern of response of vlPAG units at CS+ onset is mainly unaffected by pharmacological block of MCN during consolidation, but the temporally precise activity at CS+ offset is disrupted during EE, providing additional evidence that the two responses are independent of one another.

## The effect of temporary MCN inactivation during fear consolidation on behaviour

### Effects on freezing behaviour

To assess whether muscimol infusions into the MCN during consolidation had an effect on subsequent expression of fear-related freezing behaviour, a range of measures were taken during extinction

training: % freezing during presentation of the CS+ and inter-tone interval (ITI), extinction rate during the CS+ and ITI, and the duration of freezing and movement epochs.

During presentation of the CS+, both control and muscimol animals showed extinction of the conditioned freezing response between EE vs. LE (*Figure 5A*; control, EE freezing = 74.11% ± 3.7% vs. LE freezing = 31.1% ± 7.2%, t(9) = 9.730, p<0.0001, n = 10 animals; muscimol, EE freezing = 74.17% ± 6.1% vs. LE freezing = 36.0% ± 10.7%, t(7) = 5.404, p=0.001, n = 8 animals; paired *t*-tests; for trial-by-trial variation, see *Figure 5—figure supplement 1A*). No significant differences were detected between control and muscimol animals in the percentage of time freezing during EE or LE (*Figure 5A*; EE, t(16) = 0.0091, p=0.9928; LE, t(16) = 0.3937, p=0.579; unpaired *t*-test), nor was there a difference detected in rate of extinction of freezing (*Figure 5—figure supplement 2A*; control, –1.6 ± 1.9; muscimol, –2.4 ± 1.2; t(16) = 1.00, p=0.332, unpaired *t*-test). In terms of freezing behaviour during the ITI, both control and muscimol animals showed extinction learning similar to that found during presentation of the CS+ (*Figure 5B*; EE vs. LE: control, t(9) = 6.881, p<0.0001; muscimol, t(7) = 3.952, p=0.006, paired *t*-tests; for trial-by-trial variation, see *Figure 5—figure supplement 1B*). There was no significant difference in muscimol animals by comparison to controls for either EE or LE (*Figure 5D*; EE, t(16) = 0.048, p=0.963; LE, t(16) = 0.902, p=0.929; unpaired *t*-tests). Nor was there a significant difference in the rate of extinction between groups (*Figure 5—figure supplement 2B*; t(16) = 0.336, p=0.742, unpaired *t*-test).

For baseline behaviour monitored just before extinction training (see Materials and methods), there was no difference in the duration of freezing or movement epochs between control and muscimol animals (freezing duration for control 3.1 (1.6, 6.5) s, median ± interquartile range [IQR], n = 8 and muscimol 4.5 (3.6, 14.7) s, median ± IQR, n = 8, U = 19, p=0.195, Mann–Whitney test; movement duration for control 7.3 (3.5,14.8) s, median ± IQR, n = 10, and muscimol 7.7 (1.7, 27.12) s, median ± IQR, n = 8, U = 38, p=0.877, Mann–Whitney test). However, upon presentation of the CS+, there was a significant increase in the duration of freezing epochs in muscimol compared with control animals in both EE and LE (*Figure 5C*; EE: control 9.2 (2.8, 14.7) s, median ± IQR, n = 10; muscimol 26.45 (14.5, 82.2) s, median ± IQR, n = 8, U = 13, p=0.014, Mann–Whitney test; LE: control 3.4 (2.3, 7.0) s, median ± IQR, n = 9; muscimol 9.1 (5.6, 23.5) s, median ± IQR, n = 8, U = 13, p=0.026, Mann–Whitney test). There was also a significant increase in movement epochs in EE in muscimol by comparison to control animals, but there was no significant difference in LE (*Figure 5D*; EE: control 1.6 (1.4, 2.0) s, median ± IQR, n = 10; muscimol 4.09 (2.4, 6.8) s, median ± IQR, n = 8, U = 8, p=0.003, Mann–Whitney test; LE: control 4 (2.1, 11.0) s, median ± IQR, n = 10; muscimol 8.5 (3.0, 16.02) s, median ± IQR, n = 8, U = 29, p=0.36, Mann–Whitney test).

As might be expected during extinction training, from EE to LE the duration of freezing epochs significantly decreased in both control and muscimol animals (*Figure 5C*; control, p=0.039; muscimol, p=0.008, Wilcoxon test), and there was a corresponding increase in the duration of movement epochs. However, the latter difference was statistically significant for controls (*Figure 5D*; p=0.033, Wilcoxon test) but not for muscimol-treated animals (p=0.250, Wilcoxon test). For individual measures of both freezing and movement duration, see *Figure 5—figure supplement 1C*. Taken altogether, these results therefore indicate that blocking MCN activity during consolidation does not alter the overall proportion of time spent freezing during subsequent extinction training, but does increase the duration of individual freezing bouts, that is, freezing bouts are longer but fewer in number as there are also longer bouts of movement.

To investigate the extent to which changes in unit activity correlate with fear behaviour during extinction training, the percentage of time spent freezing was compared to the average integrated area of unit responses after CS+ onset and also for the ITI following CS+ offset. Consistent with previous reports (*Ozawa et al., 2016*; *Wright and McDannald, 2019*), at CS+ onset there was a significant positive correlation in both control and muscimol groups between unit response and percentage time spent freezing for individual animals (*Figure 5E*; control, $r_{rm}$ = 0.38, p=0.048, n = 7 animals; muscimol, $r_{rm}$ = 0.58, p=0.038, n = 3 animals; repeated-measures correlation). A significant positive correlation was also present between unit response at CS+ offset and percentage time spent freezing during ITI for both control and muscimol animals (*Figure 5F*; control, $r_{rm}$ = 0.55, p=0.002, n = 7 animals; muscimol, $r_{rm}$ = 0.69, p=0.008, n = 3 animals; repeated-measures correlation).

To investigate whether the extent of change in freezing behaviour and unit response during extinction training were related, a comparison was made between the absolute change in freezing for each

animal with the mean absolute change in unit response. No significant relationship was found between freezing and unit response area at CS+ onset, nor for freezing during the ITI and unit response area at CS+ offset (*Figure 5—figure supplement 2C and D*). Taken together, these results therefore suggest that the magnitude of vlPAG unit responses at both onset and offset of a conditioned tone reflects general fear state within an individual animal, but does not predict which individuals will have a stronger freezing response overall.

## Effects on other defence-related behaviour

To investigate whether MCN inactivation during consolidation had an effect on other defence-related behaviours, rearing activity and USVs were monitored during presentation of the CS+ and ITI during extinction training. However, by comparison to controls, no statistically significant effects were observed in either type of behaviour in muscimol animals (*Figure 5—figure supplement 2E–H*).

## Modulation of direct MCN to vlPAG projection during fear acquisition and early consolidation

Given that during extinction training (i) population activity in MCN resembles changes in vlPAG population and unit activity, and (ii) the finding that global inactivation of MCN during consolidation can lead to changes in the fear network that disrupt encoding in vlPAG, it was of interest to determine whether the behavioural effects described above were dependent on a direct projection between the two brain structures (*Whiteside and Snider, 1953*; *Teune et al., 2000*; *Vaaga et al., 2020*; *Frontera et al., 2020*). As a first step, a direct anatomical connection was investigated by injecting a fluorescently tagged anterograde virus into the MCN (n = 7 rats, *Figure 6—figure supplement 1*). In every case, terminal projections in the PAG were primarily localised to its ventrolateral region on the contralateral side.

To investigate the function of this direct MCN-PAG projection, a viral vector encoding DREADD hM4D(Gi) (n = 10, AAV-hSyn-hM4D(Gi)-mCherry, termed DREADD, see Materials and methods) or a control virus (n = 9, pAAV-hSyn-EGFP) was injected bilaterally into the MCN. Terminal projections containing either virus were targeted by implanted cannulae in the PAG to deliver clozapine N-oxide (CNO) (for histological reconstruction of cannulae loci, see *Figure 6—figure supplement 2A*). Note that while in the majority of our cases the infusion was centred on vlPAG, in keeping with studies of this type we cannot exclude spread to neighbouring areas of the midbrain.

To estimate the population of neurons that were manipulated in our viral vector experiments, in four animals we injected a retrograde tracer into vlPAG and the DREADD virus into MCN, and counted the proportion of double-labelled neurons in MCN (*Figure 6—figure supplement 3A*). Double-labelled neurons in MCN represented on average 70% ± 10.2% and 28% ± 8.3 when expressed relative to the PAG and DREADD single-labelled populations, respectively (*Figure 6—figure supplement 3B and C*). These anatomical results therefore suggest that MCN-PAG projection neurons were likely to be transfected with DREADD in our experiments.

To determine the effect of MCN-PAG pathway modulation on conditioned fear behaviours, CNO was infused 15 min (*Stachniak et al., 2014*; *Jendryka et al., 2019*) before acquisition training to modify the direct MCN-PAG pathway during acquisition and early consolidation (*Figure 6A*). During acquisition, there was no significant difference in the percentage of freezing between control versus DREADD animals during CS-US paired presentations (*Figure 6B*, upper panel, two-way repeated-measures ANOVA; time, $F(4.45, 75.64) = 16.63$, p<0.0001; virus, $F(1, 17) = 0.089$, p=0.769) nor during the ITI (*Figure 6B*, lower panel, two-way repeated-measures ANOVA; time, $F(3.834, 65.19) = 17.82$, p<0.0001; virus, $F(1, 17) = 2.339$, p=0.144). However, comparison of the mean total number of USVs per animal in DREADD versus control groups across all acquisition trials showed that during the ITI there was a significant reduction in the number of USVs (*Figure 6C*, lower panel, a decrease of 36%, control 34 (0.0 201.5) s, median ± IQR, DREADD 0 (0.0 29.25) s, median ± IQR, U = 23, p=0.032, Mann–Whitney test, one-tailed). This difference was not apparent during CS-US presentations (*Figure 6C*, upper panel, control 0 (0.0 10.0) s, median ± IQR, DREADD 0 (0.0 4.5) s, median ± IQR, U = 39, p=0.310, Mann–Whitney test, one-tailed).

During extinction training, control and DREADD animals showed similar levels of freezing during presentation of the CS+ (*Figure 7A*; EE t(17) = 0.469, p=0.645, LE t(17) = 0.937, p=0.362, unpaired *t*-test) and the ITI (*Figure 7B*; EE t(17) = 0.687, p=0.501, LE t(17) = 0.822, p=0.422, unpaired *t*-test);

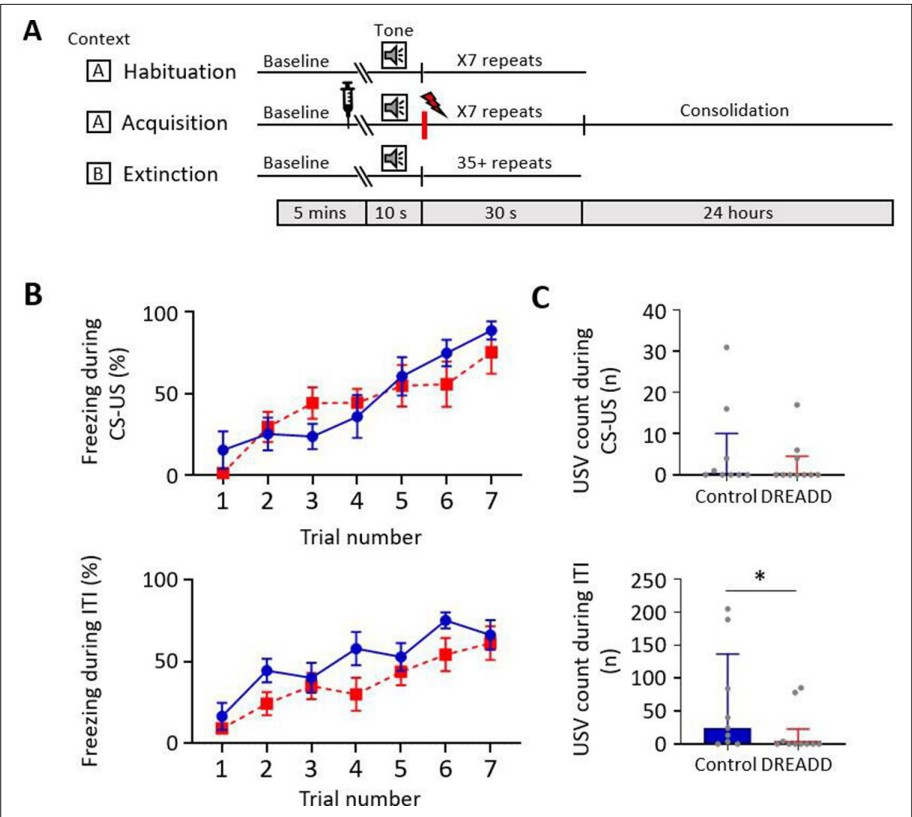

**Figure 6.** Effect of medial cerebellar nucleus-periaqueductal grey (MCN-PAG) pathway modulation on behaviour during acquisition. (**A**) Schematic timeline representing the fear conditioning protocol for the DREADDs experiment. (**B**) The effect of clozapine N-oxide (CNO) delivery into the ventrolateral PAG (vlPAG) during acquisition on freezing behaviour measured during presentation of the paired conditioned stimulus-unconditioned stimulus (CS-US) (upper plot), and the inter-tone interval (ITI) (lower plot). Blue plot, control animals (n = 9 rats); red plot, DREADD (hM4D(Gi)) animals, n = 10 rats; data points mean ± SEM. (**C**) The number of ultrasonic vocalisations (USVs) recorded in control versus DREADD animals during CS-US presentation (upper panel) and during the ITI (lower panel); bars show median ± IQR. Mann–Whitney, one-tailed test, *p<0.05.

The online version of this article includes the following source data and figure supplement(s) for figure 6:

**Source data 1.** Numerical data to support graphs in *Figure 6*.

**Figure supplement 1.** Anatomical mapping of the medial cerebellar nucleus-periaqueductal grey (MCN-PAG) pathway.

**Figure supplement 2.** Effect of DREADDs on general motor and affective behaviour.

**Figure supplement 2—source data 1.** Numerical data to support graphs in *Figure 6—figure supplement 2*.

**Figure supplement 3.** Anatomical evaluation of DREADD transfection of medial cerebellar nucleus-periaqueductal grey (MCN-PAG) pathway.

**Figure supplement 3—source data 1.** Numerical data to support graphs in *Figure 6—figure supplement 3*.

for trial-by-trial variation, see *Figure 7—figure supplement 1A and B*. Both groups also showed similar levels of extinction learning during CS+ (control, t(8) = 3.335, p=0.010; DREADD, t(9) = 10.31, p<0.0001, paired *t*-test) and the ITI (control, t(8) = 3.950, p=0.004; DREADD, t(9) = 8.679, p<0.0001, paired *t*-tests). However, the rate of extinction during the CS+ was significantly slower in DREADD animals (*Figure 7C*; t(17) = 2.2.01, p=0.042, unpaired *t*-test), but not during the ITI (*Figure 7D*; t(17) = 1.096, p=0.229, unpaired *t*-test). The latter finding suggests that the effect is not a general disruption of the expression of freezing behaviour.

With regard to rearing behaviour, no differences were found between control and DREADD animals both within EE and LE (*Figure 7E and F*; EE CS+, t(17) = 1.058, p=0.305; EE ITI, t(17) = 0.124, p=0.903; LE CS+, t(17) = 1.134, p=0.273; LE ITI, t(17) = 0.276, p=0.786, unpaired *t*-test), nor across

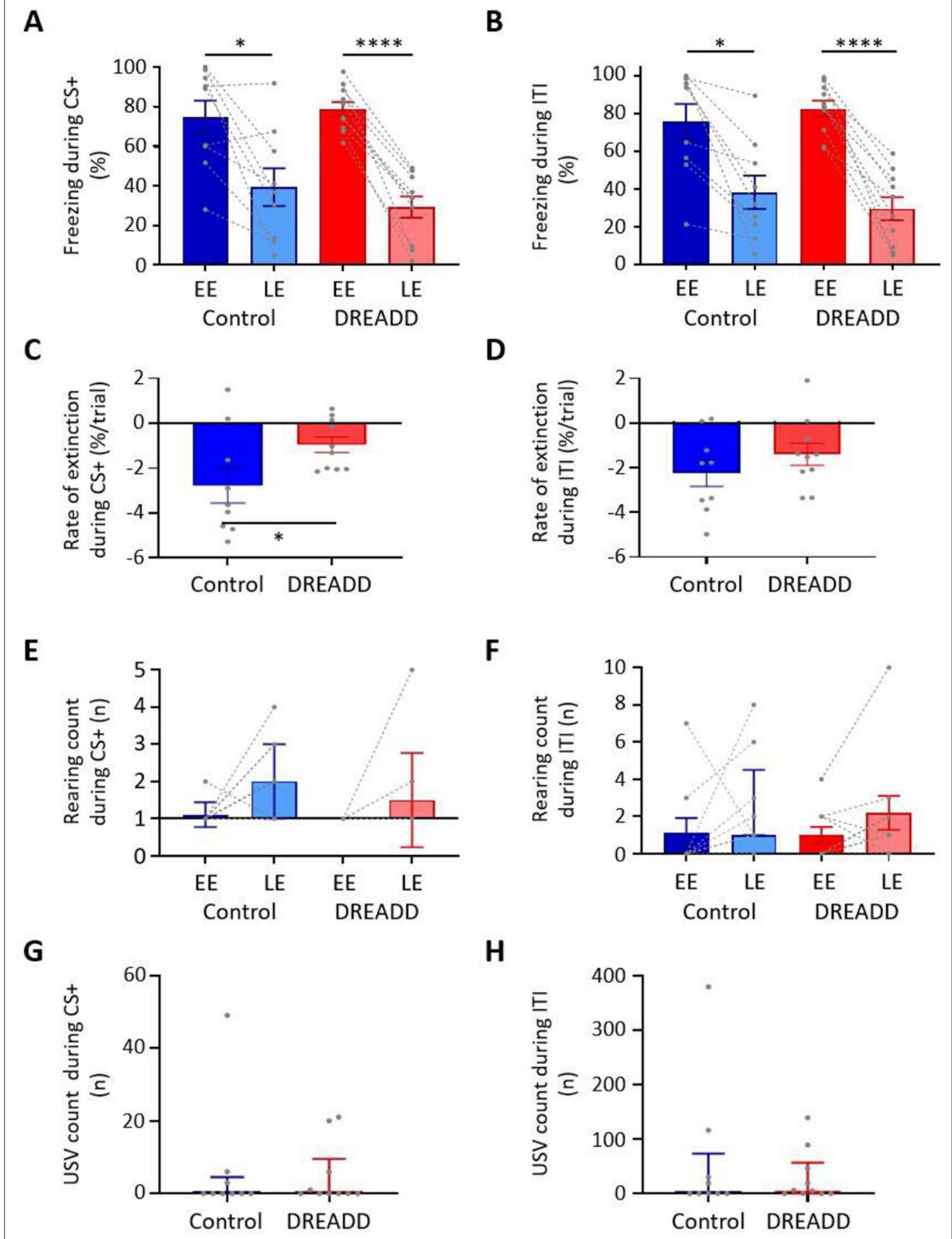

**Figure 7.** Effect of medial cerebellar nucleus-periaqueductal grey (MCN-PAG) pathway modulation on behaviour during extinction. (**A**) The percentage of total time the CS+ was presented that animals displayed freezing behaviour during early extinction (EE) (trials 1–14) or late extinction (LE) (trials 21–35) in control (n = 9 rats) versus DREADD (hM4D(Gi), n = 10 rats). Pairs of data points connected with dashed lines show the change in mean percentage freezing per animal over extinction training. Bars show group means ± SEM. ****p<0.0001, *p<0.05. (**B**) Same as (**A**) but for inter-tone interval (ITI).

*Figure 7 continued on next page*

*Figure 7 continued*

(**C**) Rate of extinction during presentation of the CS+ as measured by the change in percentage freezing over the first 21 CS+ presentations during extinction training in control (n = 9 rats) versus DREADD experiments (n = 10 rats). Individual data points show mean rate of change per animal. Bars show group means ± SEM. Unpaired *t*-test *p<0.05. (**D**) Same as (**C**) but for ITI. (**E**) The total number of rears during presentation of the CS+ during EE and LE for control (n = 9 rats) versus DREADD experiments (n = 10 rats). Pairs of data points connected with dashed lines show change in mean number of rears per animal over extinction training. Bars show group median and IQR. (**F**) Same as (**E**) but for ITI. (**G**) Total number of ultrasonic vocalisations (USVs) during presentation of the CS+ in extinction training in control (n = 9 rats) versus DREADD experiments (n = 10 rats). Individual data points show mean count per animal. Bars show group median and IQR. (**H**) Same as (**G**) but total number of USVs recorded during ITI.

The online version of this article includes the following source data and figure supplement(s) for figure 7:

**Source data 1.** Numerical data to support graphs in *Figure 7*.

**Figure supplement 1.** Freezing and ultrasonic vocalisation (USV) behaviour across extinction training.

**Figure supplement 1—source data 1.** Numerical data to support graphs in *Figure 7—figure supplement 1*.

all extinction training (*Figure 7E and F*; EE vs. LE; control CS+, t(8) = 0.610, p=0.556, control ITI, t(8) = 1.189, p=0.269; DREADD, CS+ t(9) = 1.246, p=0.244; DREADD, ITI, t(9) = 1.765, p=0.111, paired *t*-test). Similarly, there was also no significant difference in the total number of USVs per animal in DREADD versus control groups across all extinction training (*Figure 7G and H*; CS+, control 0 (0.0 4.5) s, median ± IQR, DREADD 0 (0.0 9.5) s, median ± IQR, U = 42.5, p=0.844; ITI, control 0 (0.0 73.0) s, median ± IQR, DREADD 0 (5.5 56.75) s, median ± IQR, U = 42.5, p=0.849, Mann–Whitney tests; for trial-by-trial variation, see *Figure 7—figure supplement 1C and D*).

## Control experiments for MCN-PAG intervention

To test for potential changes in anxiety and motor behaviour induced by modulation of the MCN-PAG pathway, the same DREADD and control animals studied in the fear conditioning experiments (n = 18 rats) were also tested on the following tasks after CNO infusion into vlPAG: elevated plus maze, open field, and beam walking. By comparison to controls, no statistically significant difference was found in any of these behaviours (*Figure 6—figure supplement 2B–E*) Thus, no evidence was obtained to suggest that targeted modulation of the MCN-PAG pathway causes a general deficit in motor behaviour, nor a change in anxiety levels. However, insufficient effect of the DREADD on the MCN-PAG pathway cannot be ruled out.

In all available animals (controls, n = 8; DREADD, n = 9), we therefore made an intraperitoneal injection of CNO in order to test for non-specific effects, including general action on MCN and its projections. Consistent with the muscimol experiments, in every DREADD case, the CNO produced ataxia, although this was generally less severe and shorter lasting (~1 hr) than was observed in the muscimol animals. Impairments in motor behaviour were assessed by testing the animals on the beam walking task; DREADD animals were significantly slower than control animals at traversing the beam (*Figure 6—figure supplement 2F*, t(16) = 3.192, p=0.006, unpaired *t*-test).

## Discussion

We have shown that vlPAG neurons (type 1 units) encode temporally precise information about both the onset and offset of a fear-conditioned auditory stimulus and that these two neuronal signals may be generated by independent mechanisms. This is because during extinction (i) some vlPAG units only respond to CS+ onset or only to CS+ offset; (ii) unit onset and offset responses exhibited different characteristics during extinction training; (iii) MCN inactivation during consolidation disrupted the vlPAG pattern of unit activity at CS+ offset but not onset; (iv) the latency of the ERP in vlPAG at CS+ onset was significantly shorter than the ERP recorded at CS+ offset; and (vi) the ERP recorded at CS+ onset remained similar in amplitude over extinction training while the ERP at CS+ offset showed a significant reduction.

Importantly, vlPAG units displayed little or no response to an auditory tone during habituation but displayed robust activity at tone onset and/or offset when the same tone was classically conditioned. This provides evidence that the responses were related to the associative conditioning rather than the sensory stimulus. Moreover, disruption of the temporal precision of the conditioned tone offset neural response following MCN inactivation during consolidation suggests a role of the cerebellum in the subsequent timing, but not the generation, of the vlPAG offset response. However, the possibility

remains that at other times during conditioning (acquisition and/or retrieval) the cerebellum is also involved in generating the response in vlPAG, but this requires further study.

More generally, our study applied novel approaches to address fundamental questions about the role of cerebellar-PAG interactions in survival behaviour. As such, consideration of the methods used is critical to the interpretation of the data and is included in the Materials and methods section.

## Comparison to previous behavioural studies

Previous inactivation studies in rats using TTX have indicated that the cerebellar vermal cortex is involved in the consolidation of associatively conditioned cue and context-dependent fear memories (*Sacchetti et al., 2002*; *Sacchetti et al., 2007*). This suggests that the vermal compartment of the cerebellum is involved in various aspects of fear learning. The finding in the present study that inactivation of a major output of the vermal compartment, MCN, leads to deficits in both USVs and freezing behaviour strongly supports this multiplicity of function.

The present results are also consistent with *Frontera et al., 2020*, who found in mice that inactivation of the MCN-PAG pathway during acquisition but not consolidation reduced the subsequent rate of extinction of conditioned freezing behaviour during retrieval. The present experiments in rats found that DREADD-induced modulation of the MCN-PAG pathway during acquisition slowed subsequent extinction rate, while muscimol inactivation during consolidation had no detectable effect on freezing behaviour (suggesting comparability across species). The additional finding reported here is the observation that muscimol inactivation during consolidation increases the duration of both freezing and movement bouts during retrieval of the conditioned response. This suggests that during consolidation the influence of MCN on survival circuits has an effect on the subsequent temporal profile of behavioural responses during retrieval but not the strength of the fear-related memory. We also show that modulation of the MCN-PAG pathway during acquisition reduces the expression of fear-related USVs at this time. Taken all together, this raises the possibility that the MCN regulates different aspects of survival behaviour depending on the stage of fear conditioning.

## Comparison to previous electrophysiological studies

In regard to our electrophysiological findings, similar responses to CS+ onset in vlPAG have been reported previously by ourselves (*Watson et al., 2016*) and others (*Ozawa et al., 2016*; *Wright and McDannald, 2019*), but to our knowledge no equivalent response precisely time locked to CS+ offset has been described. This may be due to differences in the characteristics of the auditory cue used for conditioning (e.g. duration, intensity, and rise-fall time of the tone; *Takahashi et al., 2004*; *Qin et al., 2007*; *Scholl et al., 2010*; *Harris et al., 2017*; *Sołyga and Barkat, 2019*), but also because detection of such responses depends on the temporal resolution used for the analysis. For example, *Watson et al., 2016* used, as is the case in many other electrophysiological studies, a 1 s bin width to visualise the patterns of vlPAG activity during CS+ presentation. Because offset responses are brief, temporally precise events, we found that a 40 ms bin width was needed to reliably capture them.

Fear-conditioned CS+ onset responses in vlPAG are thought to be generated, at least in part, as a result of preceding activity in the amygdala (*Tovote et al., 2016*; *Watson et al., 2016*) and may encode multiple aspects of fear processing, including maintenance of fear memory after extinction (*Watson et al., 2016*), prediction error coding (*Johansen et al., 2010*; *Ozawa et al., 2016*), the transmission of aversive teaching signals to the amygdala (*Johansen et al., 2010*), and threat probability (*Wright and McDannald, 2019*). Activity within vlPAG has also been correlated to the onset of freezing behaviour (*Tovote et al., 2016*; *Watson et al., 2016*), indicating that there may be several distinct neuronal populations in the PAG that are responsive to CS onset. Our results for CS+ onset unit responses are consistent with *Watson et al., 2016* as we show the phasic increase in neural activity was dependent on associative conditioning, and type 1 onset responses were generally smaller in LE by comparison to EE (i.e. were extinction sensitive) and therefore may contribute to the neural plasticity associated with extinction learning.

In terms of previous reports of CS+ offset responses in vlPAG, *Wright and McDannald, 2019* identified a distinct population of units in an auditory fear discrimination task they termed ramping units. These units were related to threat probability and also to fear output as determined by freezing behaviour. Ramping units progressively increase activity over the duration of the auditory cue presentation, reaching a peak around sound offset and ramping down in activity thereafter. This pattern

differs markedly from our type 1 offset units whose phasic activity was precisely coupled to CS+ offset. Indeed, we found no evidence of ramping-type activity in our sample of vlPAG units. This is perhaps unsurprising because the experimental paradigm of *Wright and McDannald, 2019* differed from ours in a number of important ways. In particular, they used a trace fear conditioning protocol where the aversive footshock was paired in each session with the auditory cue with a varying probability of occurrence. The closest comparison with our results is between our acquisition sessions and their trials when the probability of a footshock was 100% (their Figure 1). In our experiments, immediately after the footshock (the stimulus artefact prevented data capture during the US) we observed an increase and a subsequent progressive reduction in spike activity that resembled the change in firing after peak in their ramping units (our *Figure 1C*).

An unanswered question concerns the origin of CS+ offset responses in vlPAG. We estimate that they occur approximately 30 ms after the end of the CS+ tone. Such a delay provides ample opportunity for many possible pathways to generate them. Detecting the offset (and onset) of sensory events is a fundamental requirement of sensory processing by the CNS. Given the importance in the present experiments of the auditory system in the initial processing of the tone signals used for fear conditioning, one possibility is a route from the auditory cortex to the PAG. Both onset and offset responses have been reported in the auditory cortex elicited by tones and other sounds in a range of species (*Qin et al., 2007*; *Tian et al., 2013*; *Liu et al., 2019*). Also, changes in activity in the auditory association cortex during fear conditioning have been shown to precede the expected time of the US (*Quirk et al., 1997*). In principle, such activity occurs sufficiently in advance of our CS+ offset responses to be driving them. However, pathways from the auditory system to the PAG target its dorsolateral and lateral sectors and are associated with flight behaviour (e.g. *Wang et al., 2019*), so presumably some other pathway is responsible. For example, the medial prefrontal cortex and the bed nucleus of stria terminalis have extensive projections to the PAG (*Holstege et al., 1985*; *An et al., 1998*), and after fear conditioning both show sustained changes in activity during presentation of the CS+ (*Haufler et al., 2013*; *Gilmartin et al., 2014*).

A further important question relates to what information CS+ offset responses encode. Learning theory proposes that Pavlovian fear conditioning is instructed by an error signal that encodes the difference between actual and expected intensity of the US (*Rescorla, 1971*; *McNally et al., 2011*). The vlPAG is generally considered to provide the teaching signal that encodes prediction error to regulate synaptic plasticity in the amygdala and prefrontal cortex during fear extinction learning (*Johansen et al., 2010*; *McNally et al., 2011*; *Roy et al., 2014*; *Walker et al., 2020*; *Frontera et al., 2020*). According to the Rescorla–Wagner learning model, this teaching signal is modulated by expectation of the US – during retrieval of a fear-conditioned association, the unreinforced CS+ produces a negative prediction error signal because the US has not occurred as expected. The reliability of this prediction is reduced with successive presentations of the unreinforced CS+. If fear extinction is instructed by this error signal, then neurons encoding prediction errors would be expected to progressively decrease their CS+-induced firing rate upon repeated omission of the expected US (*McNally et al., 2011*). In the present study, the gradual reduction in CS+ type 1 offset responses in vlPAG during extinction training is entirely consistent with this proposition. However, in our trace conditioning experiment the failure of CS+ offset responses to follow the timing of the expected US would seem to argue against this, although the 1 s time interval we used may have been too short for rats to discriminate. Another possibility is the CS+ offset response is signalling saliency of the tone, but this can also be thought of as a component of generating prediction error. Clearly further studies are required, but the current findings open new avenues for investigating the role of vlPAG in encoding fear memory.

We also show that inactivation of the cerebellar output nucleus MCN during consolidation disrupts but does not abolish CS+ type 1 offset responses in vlPAG upon retrieval of the fear memory. This is in line with MCN modulating vlPAG activity (*Vaaga et al., 2020*), and advances understanding by raising the possibility that the vermal compartment of the cerebellum is involved in the timing of the memory trace, but not necessarily the origin of CS+ offset signals in vlPAG. The finding that CS+ onset and offset ERP responses occur earlier in the PAG compared with MCN would appear to provide contradictory evidence that MCN output can influence PAG activity. However, it is important to bear in mind that manipulation of the MCN-PAG pathway was carried out during consolidation, prior to retrieval. This would provide the means to modulate the subsequent timing of PAG neuronal activity, presumably through a long-term effect on survival circuits. The pattern and duration of MCN manipulation

may also be important. In mice, phasic optogenetic stimulation of the MCN-PAG pathway during CS+ offset significantly enhanced extinction learning, while tonic activation using chemogenetics had the opposite effect (*Frontera et al., 2020*), suggesting that the temporal pattern of activation of vlPAG neurons by MCN determines the effect on extinction learning.

### Functional significance

A role of MCN in temporal patterning is in good agreement with the timing hypothesis of cerebellar function (*Ivry, 1997*; *Cheron et al., 2016*; *D'Angelo, 2018*). This hypothesis proposes that the cerebellum not only regulates the timing of movements to enable coordinated behaviour and motor learning, but that this temporal regulation extends to other functions of the CNS, including perceptual tasks that require the precise timing of salient events (*Spencer and Ivry, 2013*). The present study extends this concept to the encoding of fear memory by vlPAG. Our findings suggest that the cerebellum is important for the regulation of fear memory processes at multiple timescales: at the millisecond timescale to control the neural dynamic encoding of CS+ offset within vlPAG, and at longer timescales (seconds/hours/days) to regulate the duration of freezing and movement bouts during extinction, the rate of fear extinction, and the timing of expression of fear-related behaviours – MCN indirectly influences fear-conditioned freezing behaviour during extinction training but more directly is also involved in the expression of USVs during acquisition. It is tempting to speculate that the influence of MCN on survival circuits during consolidation that leads to the temporally precise encoding of CS+ offset by vlPAG during retrieval also underlies the behavioural effects we observed during extinction, but this remains to be determined.

Cerebellar manipulations can affect emission of USVs (*Fujita et al., 2008*; *Fujita et al., 2012*; *Umeda et al., 2010*; *Fujita-Jimbo and Momoi, 2014*). Taken together with our USV results, this raises the possibility that the MCN-PAG pathway regulates the emission of USVs at a time when danger is greatest, perhaps as a warning signal to conspecifics. Our results in rats also suggest that activity in the MCN-PAG pathway during acquisition regulates the subsequent rate of fear extinction, consistent with a previous study in mice (*Frontera et al., 2020*). Outbred strains of rats have been shown to demonstrate different behavioural phenotypes during fear extinction (*Ji et al., 2018*) where a subset of animals show faster rates of extinction than others. Interruption of MCN-PAG interaction during acquisition or early consolidation may therefore contribute to an anxiety-like behavioural phenotype, with wider implications for possible neural mechanisms that underlie psychiatric disorders such as PTSD.

In summary, MCN is part of a survival circuit network that regulates the precise temporal encoding of fear memory within vlPAG and also regulates the expression of different survival behaviours depending on the phase of Pavlovian fear conditioning. Inactivation of MCN output during consolidation increases the duration of freezing and movement epochs during extinction, while a direct pathway to the vlPAG appears to be important in eliciting fear-related USVs during acquisition, and the rate of expression of freezing during EE. The cerebellum, through its interactions with the survival network, might therefore be coordinating the most appropriate behavioural response at the most appropriate time.

## Materials and methods

### Key resources table

| Reagent type (species) or resource | Designation | Source or reference | Identifiers | Additional information |
|---|---|---|---|---|
| Strain, strain background (species) | Sprague–Dawley (rat) male | Envigo | RRID:RGD_737903 | |
| Transfected construct | pAAV-CAG-tdTomato | Addgene, USA | RRID:Addgene_59462 | Adeno-associated viral vector (AAV1) |
| Transfected construct | rAVV-CAG-GFP | Addgene, USA | RRID:Addgene_37825 | Adeno-associated viral vector (AAVrg) |
| Transfected construct | pAAV-hSyn-hM4D(Gi)-mCherry | Addgene, USA | RRID:Addgene_50475 | Adeno-associated viral vector (AAV5) |
| Transfected construct | pAAV-hSyn-EGFP | Addgene, USA | RRID:Addgene_50465 | Adeno-associated viral vector (AAV5) |

*Continued on next page*

*Continued*

| Reagent type (species) or resource | Designation | Source or reference | Identifiers | Additional information |
|---|---|---|---|---|
| Chemical compound, drug | Muscimol | Sigma-Aldrich | M1523 | |
| Chemical compound, drug | Clozepine-N-oxide | Tocris Bioscience, UK | 4963 | |
| Antibody | Anti-mCherry (rabbit polyclonal) | BioVision | 5993 | (1:2000) |
| Antibody | Alexa Fluor 594 | Molecular Probes | | (1:1000) |
| Other | CerePlex μ Headstage | Blackrock Microsystems, UT | PN-9716 | |
| Software, algorithm | Blackrock Central Software Suite | Blackrock Microsystems, UT | | |
| Software, algorithm | OBS | Open Broadcaster Software; 2012–2020 | | |
| Software, algorithm | Solomon Coder | András Péter, 2019 | | |
| Software, algorithm | DeepLabCut | *Wei and Kording, 2018* | | |
| Software, algorithm | MATLAB | MathWorks | RRID:SCR_001622 | |
| Software, algorithm | Spike7 | Cambridge Electronic Design Limited | | |
| Software, algorithm | NeuroExplorer | Plexon | | |
| Software, algorithm | RStudio | RStudio, USA | | |
| Software, algorithm | Prism 9 | GraphPad, USA | | |

## Animals

All animal procedures were performed in accordance with the UK Animals (Scientific Procedures) Act of 1986 and were approved by the University of Bristol Animal Welfare and Ethical Review Body (PPL number: PA26B438F). A total of 47 adult male Sprague–Dawley rats (280–400 g; Harlan Laboratories) were used in this study. They were housed under normal environmental conditions in a normal 12 hr dark/light cycle and provided with food and water ad libitum. Animals were single housed after surgery to prevent damage to implants.

## Surgical procedures for chronic implants

Rats were anaesthetised initially with gaseous isoflurane, followed by intraperitoneal injections with ketamine and medetomidine (5 mg/100 g of Narketan 10 and Domitor, Vetoquinol). Each animal was mounted in a stereotaxic frame with atraumatic ear bars, and surgery was performed under aseptic conditions. Depth of anaesthesia was monitored regularly by testing corneal and paw withdrawal reflexes with supplementary doses of ketamine/medetomidine given as required. A midline scalp incision was made, and craniotomies were performed to gain access to the cerebellum and/or the PAG as required in each line of experiment. Microdrives and/or cannulae were implanted and/or viral injections were made as described below depending on the experiment. At the end of every surgery, the rat was administered the analgesic Metacam (Boehringer Ingelheim, 1 mg/kg) and the medetomidine antidote Atipamezole (Antisedan, Vetoquinol 0.1 mg i.p.). Animals were handled for 1 week prior to surgery and during recovery before any behavioural paradigms or electrophysiological recordings were undertaken.

## Electrophysiological recordings and cannulae (n = 22 rats)

(1) *Dual microdrive experiments (n = 6 rats).* Two in-house-built microdrives, designed to slot closely next to each other, were positioned over craniotomies to allow tetrodes to be independently advanced into the right MCN (11.4 mm caudal from bregma, 1 mm lateral from midline, depth of 4 mm) and contralateral vlPAG (7.5 mm caudal from bregma, 1 mm lateral from midline, depth 4.8 mm). The microdrives were attached to the skull with screws and dental acrylic cement. Each microdrive contained

3–4 tetrodes for local field potential (LFP) and single-unit recording (0.0008-inch tungsten wire 99.95% CS 500 HML, insulated with VG Bond, 20 µm inner diameter, impedance 100–400 kΩ after gold plating; California Fine Wire). (2) *Single microdrive experiments (n = 12 rats).* These implants were the same as described above except that only one microdrive was implanted to record single units from the vlPAG. In eight of these animals, infusion cannulae were implanted bilaterally to target the MCN (four with muscimol and four with saline) during delay conditioning experiments (details below). The remaining four animals were used in trace conditioning experiments. Single units were recorded from three of these animals.

Since similar electrophysiological data (see *Figure 2—figure supplement 1*) were recorded from the PAG obtained from animals with a dual microdrive (PAG and MCN, tetrodes-only controls) and those with a single microdrive (PAG, which received a saline infusion into MCN, saline controls), the results have been pooled (collectively termed 'control'). There was no significant difference between the overall firing rates across habituation, acquisition, and extinction training (*Figure 1—figure supplement 1C*; $F_{(2, 132)} = 0.3644$, p=0.7039, one-way ANOVA). Thus, it seems reasonable to assume that single-unit recording was stable over time and comparable between groups.

## Muscimol infusions

Effects of the muscimol infusion on general motor coordination were carefully monitored. Immediately after the infusion all animals (n = 8 rats) displayed mild to moderate ataxia, providing a positive control that the muscimol was disrupting cerebellar activity. The severity of the ataxia gradually reduced over several hours, and the animals were behaviourally normal after 24 hr.

## Anatomical pathway tracing (n = 7 rats)

To anterogradely map direct connections between MCN and PAG, 100 nl of an adeno-associated viral (AAV1) vector expressing tdTomato under the CAG promoter was injected unilaterally into the MCN (11.4 mm caudal from bregma, 1 mm lateral from midline, depth of 4.5 mm). pAAV-CAG-tdTomato (codon diversified) was a gift from Edward Boyden (Addgene viral prep # 59462-AAV1; http://n2t. net/addgene:59462; RRID:Addgene59462). Injections of the AAV were made following previously published methods (*Hirschberg et al., 2017*). In brief, the glass micropipette was connected to a 25 µl syringe (Hamilton, Bonaduz, Switzerland) via tubing filled with dyed mineral oil and was then backfilled with virus using a syringe driver (AL-1000, World Precision Instruments). To monitor progress of the injection, movement of the oil-vector capillary interface was monitored. Injections were made at 200 nl/min and the pipette left in situ for 5 min prior to removal. Survival time was 3 weeks prior to terminal perfusion and histology (see below).

## DREADD surgery (n = 24 rats)

For the DREADD experiments, two AAV vectors were used: a control, pAAV-hSyn-EGFP (AAV5); and active DREADD, pAAV-hSyn-hM4D(Gi)-mCherry (AAV5) (both gifts from Bryan Roth, Addgene viral prep # 50465-AAV5 and # 50475-AAV5). Using the same techniques as outlined above, animals were injected bilaterally into MCN with 350 nl of either the control (n = 12) or the DREADD (n = 12). In the same surgery, bilateral cannulae (26-gauge guide cannula, PlasticsOne) were also chronically implanted with tips (33-gauge internal) located just above the vlPAG (7.5 mm caudal from bregma, 0.8 mm lateral from midline, at a depth of 5 mm).

## Anatomical double labelling (n = 4 rats)

To assess the proportion of PAG projecting neurons transfected with the DREADD virus, four animals were injected with the DREADD virus in the MCN as described above and with a rAVV-CAG-GFP (AAVr) retrograde virus into vlPAG (7.5 mm caudal from bregma, 0.8 mm lateral from midline, at a depth of 5 mm). Survival time was 3 weeks prior to terminal perfusion and histology (see below).

## Behavioural protocols
### Auditory cued fear conditioning (n = 46 rats)

The delay conditioning paradigm was based on *Watson et al., 2016*. On day 0, all animals underwent a session of habituation to the Skinner box (context A, Med Associates Inc, St Albans, VT) to act as a baseline for analysis prior to a session of acquisition in the same context (day 1). During acquisition,

the CS (2 kHz, 10 s tone) was paired 7× with an US (0.5 s footshock, 0.75 mA) delivered at the end of the tone, except in four animals where the timing of the US was delayed 1 s after the CS (trace conditioning). In all animals, this was followed by a single session of extinction training in a different context (context B) on either day 2 (tetrodes-only animals) or day 3 (muscimol and saline control animals). During extinction, seven tone presentations (trials) were repeated in five blocks. The first two blocks (trials 1–14) were defined as early extinction (EE), when the animal was exhibiting high levels of freezing, while the last two blocks (trials 21–35) were defined as late extinction (LE), when the animals exhibited low levels of freezing.

### Balance beam (n = 18 rats)

This test was used to assess general motor coordination and balance (*Luong et al., 2011*). Animals were trained for three consecutive days to cross a 160-cm-long beam that ended on an enclosed safety platform (six traversals of the beam per day). On each day, the beam was progressively thinner in width (6 cm, 4 cm, and 2 cm). The 2 cm beam was then used for the test day. Baseline performance was recorded and then CNO was administered either i.p. or by intracranial infusion (for details, see below). After an interval of 15 min, the animal was retested on the beam. Beam balance performance was manually scored using Solomon Coder software (András Péter, 2019), measuring the time to cross the beam for the baseline and test trials.

### Open field (n = 14 rats)

This test was used to assess both general motor behaviour and anxiety levels. Animals were exposed for the first time to the circular arena on the test day (90 cm diameter, 51 cm height). They were placed at the perimeter of the arena and were allowed to explore for 10 min. Videos were recorded of exploratory behaviour for the whole session, and DeepLabCut (*Wei and Kording, 2018*) was used to track animal behaviour. The total distance travelled, and time spent in two equivalent areas of the arena (centre and a periphery) were calculated.

### Elevated plus maze (n = 18 rats)

This test was used as an additional assessment of anxiety (*Pellow et al., 1985*). Animals were placed in a plus-shaped maze, 1 m above the floor, with two open and two closed arms (10 cm wide and 50 cm length) and allowed to explore the maze for 5 min. Performance was manually scored using Solomon Encoder software (András Péter, 2019) to calculate the percentage of total time spent in the open versus closed arms.

## Data acquisition and analysis

### Electrophysiological recording

Multisite electrophysiological data were recorded using a Blackrock Microsystems (UT) data capture system synchronised with OptiTrak software. Raw data were processed offline to extract single-unit activity and LFPs (see 'Neural data analysis' section). Neural data for single units were sampled at 30 kHz and band-pass filtered between 300 Hz and 6 kHz. LFPs were extracted from the data by downsampling to 1 kHz and band-pass filtered at 1–32 Hz.

### Auditory cued fear conditioning

Video recording of animal behaviour during the fear conditioning paradigm was captured with an Opti-Trak camera and software, allowing synchronisation with neural data. Fear-related freezing behaviour was manually scored using Solomon Coder software (András Péter, 2019). Freezing behaviour was identified as periods in which the animal had an absence of movement (except those associated with respiration and eye movements; *Blanchard and Blanchard, 1969*) while typically maintaining a crouching position. Percentage of time spent freezing was calculated for each trial during CS+ presentations and during ITIs. The duration of freezing and movement epochs was also assessed for each animal during three different time periods: (i) baseline (the average of the last five freezing and the average of the last five moving epochs immediately before CS+ onset), (ii) EE (same as baseline but the five freezing and movement epochs following the first CS+ onset), and (iii) LE (same as baseline but the first five freezing and movement epochs following onset of CS+ presentation 22). To evaluate the extinction rate of freezing behaviour, the slope of the trendline of changes in freezing % over the

initial three blocks of extinction training was calculated (trials 1–21, by trial 21, 50% of control animals reached extinction). Rearing activity was counted as the number of events in which the animal moved from the floor to standing upright on its rear limbs. The number of rearing events during CS+ presentations and ITIs was counted.

### USV recordings

USVs emitted at 22 kHz were recorded using an ANL-940-1 Ultrasonic Microphone and Amplifier (Med Associates, Inc) connected to the Blackrock Microsystems. Although USVs were recorded as an aliased signal (the maximum sampling rate of the recording system was 30 kHz, while the optimal sampling rate was 44 kHz), we were able to identify USV events. For analysis, USVs were visualised using Spike7 software (Cambridge Electronic Design Limited) and individual USV emissions manually identified and the total number counted during each recording session.

## Control behavioural tests

Beam balance, open field, and elevated plus maze tests were recorded via standard webcams linked to OBS (Open Broadcaster Software; 2012–2020).

## Inactivation experiments (n = 12)

In eight animals, muscimol (Sigma-Aldrich, 0.3 µl, at a rate of 0.3 µl/min) (or saline; n = 4 animals) was infused via indwelling cannulae at a rate of 0.3 µl/min to target MCN. The infusion was made immediately after acquisition (i.e. during early consolidation). Extinction training was carried out 48 hr after the infusion (day 3).

## DREADD experiments

A total of 24 rats were randomly assigned to either a saline control (n = 12) or a DREADD experimental group (n = 12) and coded so the experimenter was blinded. A total of 9 controls and 10 DREADD animals were included in the analysis. The remainder (five animals) were excluded because postmortem histology showed off-target cannulae placement.

Unblinding occurred once all procedures and analysis were completed. Six weeks after viral transfection (see above), animals were tested in the following behavioural paradigms: auditory cued fear conditioning, beam balance, open field, and elevated plus maze. One animal was excluded from the study after the fear conditioning test because of poor health; a further three animals were excluded from the open field analysis because of technical problems with the video recording. In every animal, a volume of 500 nl of CNO (3 µM, Tocris) was infused at a rate of 0.5 µl/min to target the PAG, 15 min prior to each behavioural test (infusion pump Harvard Apparatus, PHD 2000 Infusion). Since the effect of CNO is estimated to last 60–90 min (*Stachniak et al., 2014*; *Jendryka et al., 2019*), this meant that during fear conditioning the pathway under study was likely to be inhibited during both acquisition (which lasted about 10 min) and also a subsequent period of early consolidation.

## Physiological and behavioural verification of hM4D(Gi) receptor activation in the MCN

1. To verify the physiological activity of hM4D(Gi) receptors following MCN transfection with DREADD, we carried out whole-cell current-clamp recordings from MCN neurons in cerebellar slices obtained from n = 4 rats. Four weeks after transfection, we investigated how CNO altered spontaneous firing rate in a sample of n = 4 neurons. CNO application to the bath caused an increase in the spontaneous firing rate in three neurons (data not shown). The one additional neuron was located in the interpositus nucleus where there was no DREADD expression, and no change in activity was detected. In transfected neurons, the hM4Di-DREADD might be expected to cause a decrease in firing rate by hyperpolarising the membrane potential via G-protein inward rectifying potassium channels. However, *Locke et al., 2018* (see their Supplementary Fig. S6) reported three different patterns of response in vitro in the lateral cerebellar nucleus of mice following bath-applied CNO: an increased firing rate, a decreased firing rate, and no change in activity. An increase in firing rate, similar to our results, may be the result of disinhibition from local circuitry.

2. To examine the behavioural effect of hM4D(Gi) receptors in the CNS following MCN transfection with DREADD, in awake rats (n = 17) CNO was injected i.p. (2.5 mg/kg) 20 min before

the beam balance test. This resulted in temporary ataxia consistent with DREADD expression having a functional effect (see *Figure 6—figure supplement 2F*).

## Histology

At experimental end points, all animals were deeply anaesthetised (Euthatal, 200 mg/ml, Merial Animal Health) and terminated by transcardial perfusion (4% paraformaldehyde in 0.1 M phosphate buffer) and the brains extracted. After post-fixation, the brains were cryoprotected in 30% sucrose solution. The cerebellum was cut sagittally and the midbrain including the PAG cut coronally into sections of 40 or 60 μm thickness, respectively. To aid verification of electrode and cannula brain loci, sections were stained with cresyl violet.

For visualisation of viral expression, immunohistochemistry for anti-mCherry (1:2000, Anti-mCherry Polyclonal Antibody host rabbit, BioVision, with 5% normal horse serum; 1:1000, Alexa Fluor 594 donkey anti rabbit IgG, Molecular Probes) was performed. No signal amplification was required for the eGFP controls. Sections were visualised on an Axioskop 2 Plus microscope (Zeiss) and photomicrographs captured using AxioVision software, or with a widefield microscope (Leica DMI6000, with Leica DFC365FX camera and Leica LASX live cell imaging workstation software).

## Neural data analysis

### Spike sorting

Offline spike sorting was carried out using MClust software in MATLAB. Clustering was classified as single unit if L ratio < 0.35, ID > 15, and < 1% of interspike intervals was >2 ms (*Schmitzer-Torbert et al., 2005*). For an example tetrode recording and its isolated spikes, see *Figure 1—figure supplement 1A and B*. The firing rate of individual units was taken from MClust and verified using NeuroExplorer.

## Data analysis of single units

For habituation, acquisition, and extinction training sessions, peri event time histograms (PETHs, 40 ms time bins) of the activity of individual units were created in NeuroExplorer, with tone onset and offset as time 0. During acquisition, the footshock caused electrical interference, so it was not possible to record unit activity during the 0.5 s period of stimulus delivery. The following analysis was performed in MATLAB. In all experimental groups, PETHs of unit activity to the unreinforced conditioned tone (CS+) during extinction training were constructed for individual units over all trials (1–35) and also separately for EE (trials 1–14) and LE trials (21–35). PETHs were z-score normalised to a 5 s baseline recording of unit activity immediately before CS+ onset and data grouped according to response type and averaged. A significant response was defined as one or more consecutive 40 ms time bins where the z-score was ±2 SD from baseline mean in the first 500 ms following the tone. The area of the response was calculated as the trapezoidal numerical integration of the first 1 s after tone onset or offset (arbitrary units). Peak z-score was measured for each unit at the time where the average peak response occurred. Total peak number was calculated as the sum of all (non-consecutive) 40 ms time bins where the z-score was ±2 SD from baseline mean within the first 500 ms after tone onset. Any single units in EE or LE that had peaks occurring after 500 ms were excluded from this analysis. The first 500 ms period after tone onset was also divided into 100 ms time bins and the proportion of units with maximal peak response within each bin expressed as a percentage of total number of maximal peaks. Maximal peak per unit was identified as the response with the largest increase relative to baseline mean.

## Data analysis of auditory ERPs

*Auditory ERPs* were extracted by averaging LFP activity in relation to tone onset and offset using MATLAB (mean based on n = 14 trials per animal). The tetrode recording site yielding the largest mean amplitude peak to trough response was identified in each animal and used to calculate group average data of peak amplitude of LFP response recorded in the PAG and cerebellum.

## Statistical analysis

Statistical analysis and graphs were performed with GraphPad Prism 9. Data are shown as mean ± SEM, except for rearing behaviour, USVs, duration of freezing, and duration of movement, which

are plotted as median ± IQR. Paired *t*-tests or Wilcoxon test were used for within-group comparisons, while unpaired *t*-tests or Mann–Whitney test and ANOVA were used to compare groups. Differences were considered significant at p<0.05. Pearson r correlation was used to compare the absolute change in freezing per animal from EE to LE versus the absolute change in unit area response from EE to LE. To investigate the relationship within animals between unit response size and freezing activity and the relationship between MCN and vlPAG ERPs, repeated-measures correlation (using RStudio; see *Bakdash and Marusich, 2017*) was calculated between response area at tone onset and offset and percentage of time spent freezing during blocks of seven tones (i.e. block 1 = tones 1–7; block 2 = tones 8–14, etc.).

## Additional methodological considerations

Our experiments were confined to the study of adult male rats. Gender differences are an important issue, and an increasing body of evidence suggests that there are neurobiological differences between male and female rodents in fear processing (*Hurley and Adams, 2008*; *Bangasser and Cuarenta, 2021*). The extent to which such differences influenced our findings remains an open question for future study.

With regard to interventionist approaches, our experiments included muscimol to block MCN activity during consolidation and this induced ataxia. It is possible that the stress and anxiety of the ataxia may have interfered with the newly formed fear memory, leading to the electrophysiological and behavioural effects we observed in extinction training. However, we consider muscimol-induced fear/stress is unlikely to fully explain our findings for the following reasons:

1. The ataxia induced by the muscimol infusion into MCN during consolidation ranged across animals from mild to moderate. There was no difference between these animals and the amount of conditioned freezing displayed during retrieval, suggesting that fear learning was not related to the severity of the ataxia. More generally, we detected no change in the proportion of total time spent freezing during retrieval, suggesting that the ataxia during consolidation had little or no effect on fear learning.
2. The effect of muscimol infusion into MCN was specific to PAG encoding of CS+ tone offset and not CS+ tone onset. If our findings were due to a generalised disruption of fear learning because of fear/stress during consolidation, then it might be expected that this would affect both onset and offset response.
3. In relation to anxiety, *Oksztel et al., 2002* found that systemic administration of muscimol in rats causes a general decrease in motility but has no effect on anxiety levels. And in terms of stress, previous studies *Brinks et al., 2009*; *Uwaya et al., 2016*; *Pietersen et al., 2016*; *Lesuis et al., 2018* have reported a correlation between the amount of freezing and corticosterone levels. Given that the amount of conditioned freezing behaviour was unaffected in our experiments, this suggests that stress is also unlikely to fully explain our findings.

An additional methodological consideration is that we used muscimol to abolish all outflow from MCN (which will include both direct and indirect pathways to the PAG), while we used the DREADD method to specifically modulate the direct pathway from MCN to vlPAG. Our experimental design therefore allowed a comparison between blocking all MCN pathways and a specific pathway targeting vlPAG. The muscimol was delivered during consolidation, while in the DREADD experiments CNO was delivered just prior to acquisition so was likely to include both a period during and after acquisition (i.e. to also include early consolidation; *Jendryka et al., 2019*). This means that there was likely to be some overlap in the time course of action of the two types of manipulation (i.e. effects on early consolidation for both DREADD and muscimol). The DREADD method has recently been used in mice at different time points during fear conditioning, including the consolidation period after acquisition (*Frontera et al., 2020*). Like our muscimol results, no effects on the proportion of total time spent freezing during extinction were reported, suggesting that the two methods are comparable. This also suggests that the behavioural effects we found in our DREADD experiments are likely to be related more to an action on associative memory processes during acquisition rather than during consolidation.

Finally, in terms of our finding that the emission of USVs was highly variable between animals, several studies have reported that only about half of the animals emit 22 kHz USVs during fear conditioning (*Wöhr et al., 2005*; *Borta et al., 2006*; *Wöhr and Schwarting, 2008*; *Schwarting and Wöhr,*

*2012*). The factors that underlie this variability are unknown, although they may include anxiety state, social experience, and status (*Hegoburu et al., 2011*; *Schwarting and Wöhr, 2012*).

## Acknowledgements

We gratefully acknowledge Ms Rachel Bissett for her expert help with histological processing, Cristiana Iosif for her help with the current clamp recordings, and the Wolfson Bioimaging Facility for their support and assistance. We would like to thank Bryan Roth and Edward Boyden for supplying the viral vectors used in this work. This work was supported by BBSRC grant BB/MO19616/1 and a Wellcome Trust PhD studentship 203775/Z/16/Z.

## Additional information

### Funding

| Funder | Grant reference number | Author |
| --- | --- | --- |
| Biotechnology and Biological Sciences Research Council | BB/MO19616/1 | Charlotte Lawrenson Bridget M Lumb Richard Apps |
| Wellcome Trust | 203775/Z/16/Z | Elena Paci |

The funders had no role in study design, data collection and interpretation, or the decision to submit the work for publication.

### Author contributions

Charlotte Lawrenson, Conceptualization, Data curation, Formal analysis, Funding acquisition, Investigation, Methodology, Writing – original draft, Writing – review and editing; Elena Paci, Conceptualization, Data curation, Formal analysis, Funding acquisition, Investigation, Methodology, Writing – review and editing; Jasmine Pickford, Methodology, Writing – review and editing; Robert AR Drake, Conceptualization, Methodology, Resources, Supervision, Writing – review and editing; Bridget M Lumb, Richard Apps, Conceptualization, Funding acquisition, Methodology, Project administration, Resources, Supervision, Writing – review and editing

### Author ORCIDs

Charlotte Lawrenson http://orcid.org/0000-0003-4778-4677
Elena Paci http://orcid.org/0000-0002-6202-2825
Robert AR Drake http://orcid.org/0000-0003-2381-7198
Bridget M Lumb http://orcid.org/0000-0002-0268-6419

### Ethics

All animal procedures were performed in accordance with the UK Animals (Scientific Procedures) Act of 1986 and were approved by the University of Bristol Animal Welfare and Ethical Review Body (PPL number: PA26B438F).

### Decision letter and Author response

Decision letter https://doi.org/10.7554/eLife.76278.sa1
Author response https://doi.org/10.7554/eLife.76278.sa2

## Additional files

### Supplementary files
• Transparent reporting form

### Data availability
Source Data file contains the numerical data used to generate the figures and perform the statistics for all main and figure supplements included in the manuscript.

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
