## [Editor Report]

This study describes interactions between the cerebellum and periaqueductal grey during fear conditioning behavior in rats. The authors have used a combination of electrophysiology, behavioral paradigms, and DREADDs to uncover critical circuits that expand the role of the cerebellum beyond motor function. The results have far reaching implications as they add new context for how inter-regional connections drive complex behaviors and they will likely stimulate new ideas on important brain circuits that cause defects in neurological and neuropsychiatric diseases.

---

## [Decision Letter]

**Decision letter after peer review:**

[Editors’ note: the authors submitted for reconsideration following the decision after peer review. What follows is the decision letter after the first round of review.]

Thank you for submitting your work entitled "Cerebellum controls timing of periaqueductal grey encoding of fear memory and expression of fear conditioned behaviour" for consideration by *eLife*. Your article has been reviewed by 3 peer reviewers, including Roy V Sillitoe as the Reviewing Editor and Reviewer #1, and the evaluation has been overseen by a Senior Editor.

We are sorry to say that, after consultation with the reviewers, we have decided that your work, in its current form, will not be considered further for publication by *eLife*.

While all three Reviewers found the direction of the research and the results promising, there were a number of concerns raised. In a number of instances there is a requirement for better controls. In addition, clarification and deeper analyses were suggested for the electrophysiology and behavior experiments. The Reviewers felt that there was a dis-link between the reported histology data of where the accurately targeted electrodes/cannulas were placed and the robustness of the electrophysiology data. For the behavior, an increase in "n" is necessary in some cases, and a suggestion was made to verify that the aversive effect of the ataxia is not responsible of the changes in fear responses during extinction.

We would consider a revised manuscript as a new initial submission, provided you are able to well address all the comments and concerns raised by each reviewer.

*Reviewer #1:*

In this study, Lawrenson et al. have investigated the contribution of cerebellar projections to the ventral periaqueductal gray (vPAG) to fear-associated behaviors. The manuscript extends a prior, albeit incomplete, understanding of how the cerebellum mediates aversive conditioning via the PAG. In particular, the authors show that vPAG neurons respond both to conditioned stimulus onset and offset, but that there are likely 3 populations of neurons that participate in this response, Type 1, Type 1 onset, and Type 1 offset. They show that blocking cerebellar nuclear signaling with muscimol delivered to the medial cerebellar nucleus during consolidation preferentially effects the only vPAG signaling at offset. The correlate to this in terms of fear behavior was seen to be an increase in rearing during early extinction, with no changes seen in USV or freezing behaviors. The authors then extend their electrophysiology analysis by showing that at the population level, neurons in both the fastigial nucleus and the vPAG show alterations in firing amplitude at CS offset, but not onset, in late vs early extinction, further strengthening the case of involvement of the cerebellum in this process. Finally, the authors use inactivation of the fastigial to vPAG circuit during the acquisition phase using DREADD technology and show effects on freezing and USV behaviors. The manuscript does an elegant job of defining a key cerebellar pathway involved in fear and succeeds in broadening the role of the cerebellum beyond basic motor function. The main weaknesses of the results are related to the presentation of certain control data and in some cases a need for additional clarity in how the data were interpreted and presented.

(1) Experimental design and paradigm: Please include the timing of both DREADD injection and muscimol injection in the experimental design schematic in Figure 1a. Alongside this, there is some concern regarding the apparent break required in the timing of muscimol injection and washout in the experimental paradigm. This extends the paradigm from the 3 days described in methods (day 0, 1, 2) to 5 days (day 0, 1, (48 hour muscimol delay), day 2 (actually day 4)). Please clarify.

(2) There are some discrepancies in the figure legends and the methods regarding the number of controls. In the methods it is stated that there are 4 saline injected controls, while in the figure this number is stated as 8.

(3) As the pooled control electrophysiology data (as stated on page 4 lines 5-8) reflects both the 3 and 5 day paradigm, are similar changes seen in EE vs LE in all three groups (3 day control, 5 day saline control, 5 day muscimol experimental)? Adding PETH panels similar to Figures2/3 A & B to supplementary figure 2 would be helpful in this regard.

(4) vPAG offset responses during trace conditioning: While this is a well-designed control, these data appear too preliminary in their current form. Please increase the sample size.

(5) Muscimol effect on rearing behavior: Given the very low number of rearings observed in control animals, saline controls, DREADD, and vehicle animals I am left to wonder whether muscimol itself has an effect on rearing independent of the conditioning paradigm used. Is such data published and could that be cited? If not, what is the explanation for this? Clarification of this point would be useful especially as it is the only behavioral outcome associated with the muscimol paradigm.

(6) Please discuss and justify why only male animals were used.

(7) Please discuss why there is such a large amount of variability in the control condition as shown in Figure 5B for the inter-CS-US interval graph (bottom right).

(8) What is the level of inhibition achieved with the DREADD experiments? That is, after CNO, what does inhibition actually look like at the single cell level and/or at the local population level? What is the efficiency of inhibition after drug delivery?

(9) For the supplementary figures (and it would be helpful in all the other figures as well), for example supplementary figure 1, please show the actual histology and not just the schematized data. There seems to be some ambiguity in the locations and it is much more informative of the reader has the actual data to examine alongside the functional data.

Specific items:

1. Line 14: change "rat ventral (vPAG)" to "rat ventral PAG (vPAG)"

2. Please clarify whether the vPAG is a subsection of the longitudinal columns of the PAG – some readers may not have a detailed knowledge of this structure.

3. Figure 1C: Please make the red bar translucent, it is difficult to tell whether there is a peak being obscured or not.

4. Please refrain from using qualitative language where statistics are not done. Instead refer to the actual numbers (instead of "large majority (75%, n=24/32) use "75% (n=24/32)" or simply "majority"). Similarly regarding "small proportion (18.5%)…".

*Reviewer #2:*

This is an interesting study describing the interactions between the cerebellum and periaqueductal grey (PAG) in fear conditioning behavior in rats. To this end the authors first studied the PAG cellular responses during an auditory cued fear conditioning paradigm. Then, they performed global cerebellar inactivation with muscimol after fear conditioning, during consolidation period and they revealed ataxia in all animals and no difference during extinction training except for rearing behavior. They also performed chemogenetic experiments to inhibit cerebello-PAG pathway during acquisition and early consolidation and show effects on extinction rate of fear related freezing and ultrasonic vocalizations. The main findings are (1) a change in PAG response to the offset of the CS and an increase solely in rearing (but not freezing or vocalization) during the extinction phase following cerebellar inhibition immediately after the conditioning, (2) a reduction of vocalization and rate of extinction during the extinction phase following the selective inhibition of cerebellar terminals in the PAG. The introduction and discussion are clear and well written, the experiments seem carefully performed; yet the results seem a bit descriptive in the current state of the manuscript and the different elements that compose the manuscript are still a bit too loosely connected. There are notably several weaknesses which need to be addressed:

(1) In the experiments with muscimol in MCN (page 11), the infusion induced ataxia. This is a severe condition that likely induces intense stress and involves many structures. The interaction with the fear learning could thus result from an interference between the newly-formed memory and a muscimol-induced fear/stress. An effort to control for this effect would be needed to discard this alternate cause (e.g. could muscimol infusion in the interposed nucleus induce ataxia without affecting the cerebellum-PAG pathway? According to Teune et al. 2000 -cited in the manuscript- it does not seem to project heavily to the PAG).

(2) The results of figures 2 and 3 are presented in parallel, in a way that needlessly complicates the reading. The CS+ onset and CS+ offset neuronal responses are presented together with respectively the fear responses during and between the CS in separate figures but the link between the electrophysiological and behavioral responses is correlative (see point 4 below). A better presentation would be to provide separate figures for the electrophysiology and the behavior as in the text (or change the text).

(3) The behavioral analysis with n=4 rats (muscimol) does not have much sense given the variability of the behavior(s). More animals are needed for the behavior.

(4) The presence of a correspondence between the changes in electrophysiological and behavioral responses in each animal (explored by pairwise correlations) would reinforce the observations: are the animals with the weakest extinction showing weaker changes in the electrophysiological responses? (NB: page 7 line 2, the difference between the responses is not properly documented: a difference in significance in two conditions is not equivalent to a significant difference between the conditions).

(5) The physiological relevance of the neuronal discharge at the CS offset (where a cerebellar contribution is suspected) remains undefined. The experiments for vPAG offset responses during trace conditioning (page 11) is very interesting in light of the characterization of the offset response. Yet, this result is too anecdotal (only one animal) and not fully analyzed (no quantification for the EE and LE). The manuscript would benefit greatly from a proper demonstration of this response.

6) The ERP experiment is suffering from two limitations: (1) it is unclear how the LFP relates to the neuronal activity vs afferent activity; it would be much helpful to show using the available PAG recordings how the LFP relates to the multiunit activity. (2) the number of animals is too limited for the analysis (the lack of effect on MCN onset is only based on 4 pairs of measures in which 3 pairs show a clear decrement; btw: how is it possible to have 5 animals CS+ offset but only 4 in CS+ onset?); testing the covariance of MCN and PAG ERPs for each rat could help circumvent the limited number of animals (are the inter-trial fluctuations related in the two structures ?). More animals should be added otherwise.

7) It is difficult to judge from the specificity of the action of CNO; some cannulas (sup Figure 5, -7.5 AP) seem off-target (ventral PAG rather than vlPAG, while it contains very few cerebellar terminals according to sup Figure 4B): were these experiments included in the analysis? Terminal labeling has been found in the region bordering the vlPAG: how can the authors be sure that the behavioral effects are not due to an action on these terminals outside of the PAG? The diffusion of CNO needs to be documented to interpret these experiments.

8) If I understand correctly, the extended response to the CS-offset (Figure 3B) is interpreted as poor temporal patterning (discussion p22-23); the average trace indicates longer responses to CS+ offset after muscimol during consolidation; are these responses less precisely associated to the end of the CS ? Because the traces are only presented as a grand average, it is not possible to see whether some responses become poorly correlated to the end of the CS.

*Reviewer #3:*

The authors investigated the role of projections from the medial cerebellar nucleus (MCN) to the ventral periaqueductal gray (vPAG) in a conditioned fear learning task in rats, measuring vPAG neuronal activity and behavior before and after inhibiting MCN activity or the MCN-vPAG pathway. Inactivation of the MCN and the MCN – vPAG projections both affect fear learning related neuronal activity and behavior but in different ways. MCN inactivation results in a reduced neuronal response at the end of the conditioned tone stimulus, and also more frequent rearing and more frequent ultrasonic vocalizations during extinction. Neither manipulation changed the expression of fear as measured by freezing behavior, or the rate of fear extinction. Several statements in the manuscript are not supported by the results, such as the statement that the timing of neuronal activity signaling the end of the conditioned stimulus is altered by inhibition of the MCN or its projections.

The differences between the observed effects of the two manipulations (MCN or pathway inhibition) on fear learning may be due to the differences in the method and timing of inactivation. MCN is inactivated during consolidation. The MCN vPAC pathway is inhibited during fear acquisition, with inhibition possibly extending into early consolidation. MCN inhibition is through muscimol, pathway inhibition through use of and DREADDs. These differences introduce unknown variability into the results.

There is no description of measurements to determine how effective the inhibition of the MCN or the vPAC was, but interpretation of results is under the assumption that it was complete or near complete. While a systemic injection of CNO will deliver clozapine to all DREADD receptors (and, depending on does possibly also endogenous clozapine receptors), it does not inform about the percentage of projecting axons that express the designer receptor. It would have been reassuring to see some electrophysiological proof of the success and extent of the inhibition.

The authors don't seem to have performed necessary control experiments to exclude potential effects of CNO on endogenous clozapine binding sites in the brain when using systemic CNO injections.

The authors state that MCN inhibition alters the temporal precision of vPAG offset activity. Timing as a contribution of the cerebellum to fear learning is mentioned several times. However, there is no actual data supporting a change in timing or temporal precision. The authors don't analyze the temporal precision of the offset response, only the area under the curve and peak amplitude.

In the last paragraph of the introduction the authors claim that the rate of extinction is affected by MCN inhibition. There are no results to support this. In the same paragraph the authors state that ultrasonic vocalizations (USVs) and rearing behavior during fear acquisition are influenced by MCN inputs to vPAC. However, all results for USV and rearing counts are shown for the extinction phase.

Experimental findings with an n = 1, as the author's experiment with one rat exposed to trace fear conditioning, are not acceptable.

Supplementary Figure 1 A is supposed to show electrode tracks for rats implanted with two drives for electrodes. The left cerebellar illustration shows what is declared as canular tracks in the other illustration. Was that by mistake or where cannulas inserted here?

The manuscript overstates several findings. For example, the presented findings do not imply that the cerebellum 'controls' any of these behaviors or activity patterns but rather that it is part of a larger circuit that does.

The claim that the cerebellum control aspects of timing in this task is not supported by results. The timing of the off response, which is specifically mentioned, needs to be analyzed to support or refute this claim.

The trace fear conditioning experiment should be repeated with at least two more animals (although a higher number would likely be needed).

Manipulations of MCN activity and of the MCN-vPAG pathway should occur during the same period in the fear conditioning protocol and preferentially by using the same method to ensure similar time courses.

Inhibition of the MCN and especially be confirmed electrophysiologically, at least for some examples.

Please show some example raw data of your electrophysiological recordings.

[Editors’ note: further revisions were suggested prior to acceptance, as described below.]

Thank you for resubmitting your article "Cerebellar modulation of fear behaviour and memory encoding in the PAG." for consideration by *eLife*. Your article has been reviewed by 3 peer reviewers, including Roy V Sillitoe as the Reviewing Editor and Reviewer #1, and the evaluation has been overseen by Kate Wassum as the Senior Editor. The following individual involved in review of your submission has agreed to reveal their identity: Detlef Heck (Reviewer #3).

This study describes interactions between the cerebellum and periaqueductal grey during fear conditioning behavior in rats. The authors have used a combination of electrophysiology, behavioral paradigms, and DREADDs to uncover critical circuits that expand the role of the cerebellum beyond motor function. The results have far reaching implications as they add new context for how inter-regional connections drive complex behaviors and they will likely stimulate new ideas on the circuits that cause defects in neuropsychiatric diseases.

Essential revisions:

Please see below the specific recommendations by the reviewers. In addition to the points of clarity as requested, please pay particular attention to:

1) Cases where animal numbers need to be provided.

2) References as requested.

3) Details regarding methodology.

4) Please ensure your manuscript complies with the *eLife* policies for statistical reporting: https://reviewer.elifesciences.org/author-guide/full "Report exact p-values wherever possible alongside the summary statistics (and degrees of freedom) and 95% confidence intervals. These should be reported for all key questions and not only when the p-value is less than 0.05."

Please also include a key resource table.

*Reviewer #1:*

The authors have submitted a revised version of their manuscript that tests the role of the cerebellum in fear memory. The manuscript has been given a major overhaul with revised text, revised figures, and additional data. The authors have done an excellent job in addressing the comments provided by the reviewers on the initial submission.

*Reviewer #3:*

The authors have adequately addressed all my concerns.

One small final request would be that the authors mention the results from their test of DREADD activation and the related findings by Locke et al. 2018 in the manuscript. This could just be in writing without adding a figure.

---

## [Author Response]

[Editors’ note: the authors resubmitted a revised version of the paper for consideration. What follows is the authors’ response to the first round of review.]

Reviewer #1:In this study, Lawrenson et al. have investigated the contribution of cerebellar projections to the ventral periaqueductal gray (vPAG) to fear-associated behaviors. The manuscript extends a prior, albeit incomplete, understanding of how the cerebellum mediates aversive conditioning via the PAG. In particular, the authors show that vPAG neurons respond both to conditioned stimulus onset and offset, but that there are likely 3 populations of neurons that participate in this response, Type 1, Type 1 onset, and Type 1 offset. They show that blocking cerebellar nuclear signaling with muscimol delivered to the medial cerebellar nucleus during consolidation preferentially effects the only vPAG signaling at offset. The correlate to this in terms of fear behavior was seen to be an increase in rearing during early extinction, with no changes seen in USV or freezing behaviors. The authors then extend their electrophysiology analysis by showing that at the population level, neurons in both the fastigial nucleus and the vPAG show alterations in firing amplitude at CS offset, but not onset, in late vs early extinction, further strengthening the case of involvement of the cerebellum in this process. Finally, the authors use inactivation of the fastigial to vPAG circuit during the acquisition phase using DREADD technology and show effects on freezing and USV behaviors. The manuscript does an elegant job of defining a key cerebellar pathway involved in fear and succeeds in broadening the role of the cerebellum beyond basic motor function. The main weaknesses of the results are related to the presentation of certain control data and in some cases a need for additional clarity in how the data were interpreted and presented.(1.1) Experimental design and paradigm: Please include the timing of both DREADD injection and muscimol injection in the experimental design schematic in Figure 1a.

We thank the reviewer for this helpful suggestion. However, given that Figure 1 relates to non- drug (control) experiments we do not think this is the best place to include time of muscimol injections. Instead, we have included the schematic in Figure 2 and Figure 6 to show time of muscimol and DREADD infusion respectively.

Alongside this, there is some concern regarding the apparent break required in the timing of muscimol injection and washout in the experimental paradigm. This extends the paradigm from the 3 days described in methods (day 0, 1, 2) to 5 days (day 0, 1, (48 hour muscimol delay), day 2 (actually day 4)). Please clarify.

We included a 48hr delay to be confident there was complete washout of muscimol and no residual effects on motor performance during extinction training. The experimental timeline was therefore as follows: Habituation = day 0; Acquisition = day 1; Extinction training = day 2 (tetrodes only control) or day 3 (muscimol or saline control). As suggested in point 1.3 we have compared day 2 (tetrodes only control) and day 3 (saline control) to show this does not have an effect on our results and this finding is included in Supplementary Figure 5.

(1.2) There are some discrepancies in the figure legends and the methods regarding the number of controls. In the methods it is stated that there are 4 saline injected controls, while in the figure this number is stated as 8.

We thank the reviewer for spotting this discrepancy. We have corrected the text and clarified that there are 10 control animals in total: 6 were animals with dual tetrode implants to record from PAG and cerebellum (tetrodes only control), while 4 animals had tetrode implants in the PAG and cannulae implanted in the cerebellum, in which saline was infused (saline controls).

(1.3) As the pooled control electrophysiology data (as stated on page 4 lines 5-8) reflects both the 3 and 5 day paradigm, are similar changes seen in EE vs LE in all three groups (3 day control, 5 day saline control, 5 day muscimol experimental)? Adding PETH panels similar to Figures2/3 A & B to supplementary figure 2 would be helpful in this regard.

We have added additional PETH plots to Supplementary Figure 5, showing that results for day 2 data (tetrodes only control) and day 3 (saline control) are not statistically different from each other. However, both control groups are significantly different from the muscimol group in EE (unpaired t test p = 0.038 and p = 0.011; LE, figures not shown).

(1.4) vPAG offset responses during trace conditioning: While this is a well-designed control, these data appear too preliminary in their current form. Please increase the sample size.

We have increased the sample size of offset responses in trace conditioning experiments to 12 vPAG units recorded in a total of 3 animals. Consistent with our initial findings the data show that 6 type 1 units (those with increased activity at CS+ offset), are temporally related to the offset of the CS+ during retrieval and not to the time of the expected occurrence of the US (1s after CS+ offset). We also recorded 6 units with type 4 responses (those with reduced activity at CS+ offset) and in trace conditioning these were also related to time of CS+ offset. A comparison of time to peak of response for units recorded during control versus trace conditioning showed no statistical difference. Results section has been revised to include these additional results and Supplementary Figure 6 has been updated.

(1.5) Muscimol effect on rearing behavior: Given the very low number of rearings observed in control animals, saline controls, DREADD, and vehicle animals I am left to wonder whether muscimol itself has an effect on rearing independent of the conditioning paradigm used. Is such data published and could that be cited? If not, what is the explanation for this? Clarification of this point would be useful especially as it is the only behavioral outcome associated with the muscimol paradigm.

As indicated above (1.1.2) we administered the muscimol to MCN during the consolidation phase of fear conditioning, 48hrs before extinction training and the evaluation of rearing behaviour. It therefore seems reasonable to rule out any direct action of muscimol on the rearing behaviour we studied (or any of the other behaviours we evaluated during extinction training). Previous studies of the effects of systemic administration of muscimol on rearing behaviour found this reduced the number of rearings in rats (Oksztel et al., 2002; Scattoni et al. 2003). This is the opposite to our findings, supporting the suggestion that the effect we observed was not a direct consequence of the drug administration. However, and as highlighted by reviewer 2 (point 2.3) our original behavioural analysis was based on a small sample of animals (n = 4). We have doubled the sample (total of n = 8 animals) and the effect on number of rearings during extinction training, is no longer statistically significant (Supplementary Figure 7). Given our proposal that the MCN is involved in different aspects of timing during fear conditioning (see 2.8 and 3.4 with regard to timing of neural activity) we have analyzed the larger behavioural dataset in relation to duration of freezing and movement bouts following CS+ (Figure 5 C,D). In muscimol treated animals the duration is significantly longer (on average 65.2% longer duration in EE than control). The latter additional finding is included in a new section in Results (The effect of temporary MCN inactivation during fear consolidation on behaviour (i) Effects on freezing behaviour).

(1.6) Please discuss and justify why only male animals were used.

We confined our study to males because female rats would introduce an additional variable, namely the effects of the oestrous cycle. This would have 3R implications through greatly increasing our sample size. We have included a short section in Methods (page 34, lines 2124) recognising that gender differences are an important issue that may influence our findings (Hurley et al. 2008), but that this was beyond the scope of the current work.

(1.7) Please discuss why there is such a large amount of variability in the control condition as shown in Figure 5B for the inter-CS-US interval graph (bottom right).

The emission of USVs is a highly variable behaviour in rodents. For example, several studies have reported that only about half of animals emit 22 kHz USVs during fear conditioning (Wohr et al., 2005; Borta et al., 2006; Wohr and Schwarting, 2008, 2008). The factors that underlie this variability are unknown, although they may include the anxiety state of an individual, social experience and status (Schwarting and M. Wöhr, 2012; Hegoburu et al., 2011). We have included mention of this inherent variability in Methods pages 35, line 32-3.

(1.8) What is the level of inhibition achieved with the DREADD experiments? That is, after CNO, what does inhibition actually look like at the single cell level and/or at the local population level? What is the efficiency of inhibition after drug delivery?

This is an interesting question but one that is very difficult to answer experimentally. There is good evidence from the literature that using this technique silences presynaptic terminals by inhibiting vesicle release in the local area (Mahler et al., 2014; Stachniak et al., 2014; Zhu and Roth, 2014). However, inhibition of release at terminals may cause a number of effects in the target region depending on the nature of both pre- and post-synaptic neurons. For example, reducing excitatory drive to glutamatergic neurons will reduce subsequent excitation yet reducing inputs to GABAergic neurons may enhance excitation through disinhibition (see Locke et al. 2018 for evidence of bidirectional effects). The network effects are therefore likely to be complex and are beyond the scope of our current experiments. However, our control and experimental evidence point towards a disruption of physiological signalling in regions of interest when using DREADDs. Given the difficulty of determining if the net effect of the DREADDS is inhibitory or disinhibitory we have removed ‘inhibitory’ from the text and instead refer to ‘modulation’.

See also response to reviewer (3.2) where we highlight some additional experiments we have carried out to estimate the population of neurons in the MCN-vPAG pathway that have been manipulated. And also some brain slice experiments we have undertaken showing physiological effects of hM4Di DREADDs consistent with Locke et al. (2018).

(1.9) For the supplementary figures (and it would be helpful in all the other figures as well), for example supplementary figure 1, please show the actual histology and not just the schematized data. There seems to be some ambiguity in the locations and it is much more informative of the reader has the actual data to examine alongside the functional data.

We have revised Supplementary Figure 1 to include photomicrographs of example cases. For the summary diagrams we have revised these to show the start and endpoint of individual electrode tracks where units were recorded.

Specific items:1. Line 14: change "rat ventral (vPAG)" to "rat ventral PAG (vPAG)"

This edit has been made.

2. Please clarify whether the vPAG is a subsection of the longitudinal columns of the PAG – some readers may not have a detailed knowledge of this structure.

As in our previous work (Watson et al. 2016) we have used the term vPAG to include ventral (including ventrolateral) sectors of the PAG in which our recordings were made (see histology maps point 1.9). The term ventral is used because of the inherent limitation of using multiple tetrode implants in an awake animal. This approach increases unit yield at the expense of spatial accuracy of recording site location when reconstructing individual tetrode tracks post mortem. Our recordings and manipulations were centred on the ventrolateral longitudinal column of PAG but we cannot exclude the possibility that some units were located in neighbouring regions. This point has been added in Results (page 18, line 23-25).

3. Figure 1C: Please make the red bar translucent, it is difficult to tell whether there is a peak being obscured or not.

The red stimulus bar in Figure 1C has been made translucent to make clear there were no data obtained at this time of recording because of the stimulus artefact. A comment has also been added to the figure legend to this effect.

4. Please refrain from using qualitative language where statistics are not done. Instead refer to the actual numbers (instead of "large majority (75%, n=24/32) use "75% (n=24/32)" or simply "majority"). Similarly regarding "small proportion (18.5%)…".

The manuscript has been revised throughout to remove qualitative language.

Reviewer #2:This is an interesting study describing the interactions between the cerebellum and periaqueductal grey (PAG) in fear conditioning behavior in rats. To this end the authors first studied the PAG cellular responses during an auditory cued fear conditioning paradigm. Then, they performed global cerebellar inactivation with muscimol after fear conditioning, during consolidation period and they revealed ataxia in all animals and no difference during extinction training except for rearing behavior. They also performed chemogenetic experiments to inhibit cerebello-PAG pathway during acquisition and early consolidation and show effects on extinction rate of fear related freezing and ultrasonic vocalizations. The main findings are (1) a change in PAG response to the offset of the CS and an increase solely in rearing (but not freezing or vocalization) during the extinction phase following cerebellar inhibition immediately after the conditioning, (2) a reduction of vocalization and rate of extinction during the extinction phase following the selective inhibition of cerebellar terminals in the PAG. The introduction and discussion are clear and well written, the experiments seem carefully performed; yet the results seem a bit descriptive in the current state of the manuscript and the different elements that compose the manuscript are still a bit too loosely connected. There are notably several weaknesses which need to be addressed:(2.1) In the experiments with muscimol in MCN (page 11), the infusion induced ataxia. This is a severe condition that likely induces intense stress and involves many structures. The interaction with the fear learning could thus result from an interference between the newly-formed memory and a muscimol-induced fear/stress. An effort to control for this effect would be needed to discard this alternate cause (e.g. could muscimol infusion in the interposed nucleus induce ataxia without affecting the cerebellum-PAG pathway? According to Teune et al. 2000 -cited in the manuscript- it does not seem to project heavily to the PAG).

In our lab, muscimol infusions (with similar and larger drug doses) into the interposed nucleus do not cause ataxia (consistent with Sachetti et al. 2002 who infused TTX); only infusions into MCN caused ataxia so unfortunately the control experiment proposed by the reviewer would not be an appropriate comparison.

However, we have included the following points in the Methods section (page 34, line 20) to make the case why we think muscimol induced fear/stress is unlikely to fully explain our findings.

i) The ataxia induced by the muscimol infusion into MCN during consolidation ranged across animals from moderate (animals were unable to move around their homecage, n=2) to mild (changed gait, n=6). There was no difference between these two groups in the amount of conditioned freezing displayed during retrieval, suggesting that fear learning was not related to the severity of the ataxia. More generally, we detected no change in conditioned freezing behaviour on retrieval, suggesting that the ataxia during consolidation had little or no effect on fear learning (see also 1.5 above).

ii) The effect of muscimol infusion into MCN was specific to PAG encoding of CS tone offset and not CS onset. If our findings were due to a generalized disruption of fear learning because of fear/stress during consolidation, then it might be expected that this would affect both onset and offset responses. It is also noteworthy that stress levels determined by corticosterone are related to tone onset (Born et al., 1989) so it might be expected that this neural response would be sensitive to any changes in stress caused by ataxia.

iii) In relation to anxiety, Oskel et al. (2003) found that systemic administration of muscimol in rats causes a general decrease in motility but has no effect on anxiety levels. And in terms of stress, previous studies (Brinks et al., 2009; Pietersen et al., 2006; Lesuis et al., 2018; Uwaya et al., 2016) have reported a correlation between freezing and corticosterone levels. Given that conditioned freezing behaviour was unaffected in our experiments, this suggests that stress is also unlikely to fully explain our findings.

We have consulted with neuroendocrine experts and carefully considered additional experiments to monitor stress levels (e.g. the possibility of monitoring changes in corticosterone in muscimol versus saline infused animals during consolidation of fear learning), but this would require a large number of additional animals and in any case such experiments would not provide a definitive result.

(2.2) The results of figures 2 and 3 are presented in parallel, in a way that needlessly complicates the reading. The CS+ onset and CS+ offset neuronal responses are presented together with respectively the fear responses during and between the CS in separate figures but the link between the electrophysiological and behavioral responses is correlative (see point 4 below). A better presentation would be to provide separate figures for the electrophysiology and the behavior as in the text (or change the text).

We have revised the figures to show the electrophysiological and behavioural results separately.

(2.3) The behavioral analysis with n=4 rats (muscimol) does not have much sense given the variability of the behavior(s). More animals are needed for the behavior.

We have increased the sample size for the behavioural experiments to a total of n = 8 muscimol animals. The relevant text, stats and graphs have been updated (see also point 1.5).

(2.4) The presence of a correspondence between the changes in electrophysiological and behavioral responses in each animal (explored by pairwise correlations) would reinforce the observations: are the animals with the weakest extinction showing weaker changes in the electrophysiological responses? (NB: page 7 line 2, the difference between the responses is not properly documented: a difference in significance in two conditions is not equivalent to a significant difference between the conditions).

We have carried out a repeated measures correlation analysis (a more accurate statistical measure than pairwise comparisons as it controls for repeated observations within the same animal over time. See Bakdash and Marusich, Repeated measures correlation. *Front Psychol*, 2017) of electrophysiological response size (integrated area) at onset and offset of conditioned tone as a function of % freezing for control and muscimol experiments. Within animals there was a positive relationship, with the largest electrophysiological response size related to the largest proportion of conditioned freezing. These data are included in a revised Figure 5 (panels E,F).

Although the findings above show a relationship within animals between freezing and the electrophysiological response, we found no relationship between animals. This was tested as suggested by the reviewer by comparing the change in freezing with the change in electrophysiological response between animals using Pearson r correlation. These data are now shown in Supplementary Figure 7 C,D.

(2.5) The physiological relevance of the neuronal discharge at the CS offset (where a cerebellar contribution is suspected) remains undefined. The experiments for vPAG offset responses during trace conditioning (page 11) is very interesting in light of the characterization of the offset response. Yet, this result is too anecdotal (only one animal) and not fully analyzed (no quantification for the EE and LE). The manuscript would benefit greatly from a proper demonstration of this response.

We have increased the dataset for trace conditioning to a total of 12 vPAG units from 3 animals and quantified the findings. See point 1.4 above.

(2.6) The ERP experiment is suffering from two limitations: (1) it is unclear how the LFP relates to the neuronal activity vs afferent activity; it would be much helpful to show using the available PAG recordings how the LFP relates to the multiunit activity.

We have included a Supplementary Figure 4 to show an example case that is typical of the available data, showing the relationship between the ERP at offset and multiunit activity in PAG. This shows that the field waveform and duration of spike activity are broadly coincident.

(2.6.2) The number of animals is too limited for the analysis (the lack of effect on MCN onset is only based on 4 pairs of measures in which 3 pairs show a clear decrement; btw: how is it possible to have 5 animals CS+ offset but only 4 in CS+ onset?); testing the covariance of MCN and PAG ERPs for each rat could help circumvent the limited number of animals (are the inter-trial fluctuations related in the two structures ?). More animals should be added otherwise.

We have increased the number of ERP animals to 6 and the results updated. We have tested the covariance, with a repeated measures correlation, between MCN and PAG ERPs during extinction training (Figure 3E,F). In both cases there was a significant positive correlation. (page10, line 27-28).

(2.7) It is difficult to judge from the specificity of the action of CNO; some cannulas (sup Figure 5, -7.5 AP) seem off-target (ventral PAG rather than vlPAG, while it contains very few cerebellar terminals according to sup Figure 4B): were these experiments included in the analysis? Terminal labeling has been found in the region bordering the vlPAG: how can the authors be sure that the behavioral effects are not due to an action on these terminals outside of the PAG? The diffusion of CNO needs to be documented to interpret these experiments.

In the majority of our cases the infusion was centred on vlPAG, but in keeping with studies of this type we cannot exclude spread to neighbouring areas. This limitation is noted in Results (p18, line 23-25). Based on other experiments (unpublished data) and previously published reports (Barker et al., 2017) we estimate spread to be ~0.45 mm^3^, extending from -6.5 to 8.5 anterior-posterior and up to 2.5 laterally. In terms of being off-target we believe the reviewer is referring to one cannula that was centrally located – in this case spread of infusate would include vlPAG. Also infusions were bilateral and the other cannula of the pair in this case was on target.

(2.8) If I understand correctly, the extended response to the CS-offset (Figure 3B) is interpreted as poor temporal patterning (discussion p22-23); the average trace indicates longer responses to CS+ offset after muscimol during consolidation; are these responses less precisely associated to the end of the CS ? Because the traces are only presented as a grand average, it is not possible to see whether some responses become poorly correlated to the end of the CS.

We thank the reviewer for this helpful point. In a new Figure 4 we now include individual data for all available units as violin plots which show for control and muscimol treated animals the number of significant (≥ 2SD from baseline) peaks in neural activity in the 500ms after CS+ onset and CS+ offset for EE and LE. In control animals at CS+ offset during EE there was on average 1 peak compared to 3 peaks in muscimol animals (Figure 4C, p<0.0001). Also, the proportion of peaks in different 100 ms time bins is significantly different (Figure 4D). For control animals over 80% of peaks occurred in the 200 ms time period immediately following CS+ offset, while the corresponding value is 20% for muscimol treated animals (p<0.01, Chi-square).

Reviewer #3:The authors investigated the role of projections from the medial cerebellar nucleus (MCN) to the ventral periaqueductal gray (vPAG) in a conditioned fear learning task in rats, measuring vPAG neuronal activity and behavior before and after inhibiting MCN activity or the MCN-vPAG pathway. Inactivation of the MCN and the MCN – vPAG projections both affect fear learning related neuronal activity and behavior but in different ways. MCN inactivation results in a reduced neuronal response at the end of the conditioned tone stimulus, and also more frequent rearing and more frequent ultrasonic vocalizations during extinction. Neither manipulation changed the expression of fear as measured by freezing behavior, or the rate of fear extinction. Several statements in the manuscript are not supported by the results, such as the statement that the timing of neuronal activity signaling the end of the conditioned stimulus is altered by inhibition of the MCN or its projections.3.1. The differences between the observed effects of the two manipulations (MCN or pathway inhibition) on fear learning may be due to the differences in the method and timing of inactivation. MCN is inactivated during consolidation. The MCN vPAC pathway is inhibited during fear acquisition, with inhibition possibly extending into early consolidation. MCN inhibition is through muscimol, pathway inhibition through use of and DREADDs. These differences introduce unknown variability into the results. Manipulations of MCN activity and of the MCN-vPAG pathway should occur during the same period in the fear conditioning protocol and preferentially by using the same method to ensure similar time courses.

Our experiments included muscimol to abolish all outflow from MCN (which will include both direct and indirect pathways to the PAG) and the DREADD method to specifically inhibit the direct pathway from MCN to PAG. This experimental design therefore allowed a comparison between blocking all MCN pathways and a specific one to PAG. As stated in the Introduction we focused on effects during retrieval and extinction of fear conditioned responses because of the potential clinical relevance to conditions such as PTSD and chronic pain.

The reviewer is correct to note that the two types of manipulation were given at different time points (prior to and after acquisition for DREADD activation and muscimol respectively) and this difference is noted in the Methods (p28, line 21-31).

We chose muscimol for our initial manipulation during consolidation because the duration of its effects last longer than DREADDs activation. In our case for Muscimol the motor effects were observed for 3-6 hours and other experiments in the literature have found behavioural effects lasting 5+ hours (e.g. Hikosaka et al., 1985) meaning we effect a longer time period of consolidation. DREADD activation in comparison is for a relatively short amount of time <60mins (Stachniak et al., 2014; Jendryka et al., 2019).

We did not deliver muscimol to MCN during acquisition because of the confounding effects of the drug on movement. There are also significant welfare concerns of delivering aversive stimuli to animals with compromised mobility.

However, and as noted by the reviewer, the activation of DREADDs by CNO was likely to include both a period during and after acquisition (i.e. to also include early consolidation, Jendryka et al., 2019). This means there was likely to be some overlap in time course of action of the two types of manipulation (i.e. effects on early consolidation for both DREADDS and muscimol). The DREADD method has recently been used in mice at different time points during fear conditioning (Frontera et al., 2020), including the consolidation period after acquisition. Like our muscimol results no effects on fear conditioned freezing behaviour were reported, suggesting that the two methods are comparable and that the effects of the DREADDs are related mainly to an action on associative memory processes during acquisition. Given the comprehensive results on freezing behaviour published by Frontera et al., (2020) we do not think it is justified to repeat their experiments which are considered in the Discussion (see p23, line 29).

3.2. There is no description of measurements to determine how effective the inhibition of the MCN or the vPAC was, but interpretation of results is under the assumption that it was complete or near complete. While a systemic injection of CNO will deliver clozapine to all DREADD receptors (and, depending on does possibly also endogenous clozapine receptors), it does not inform about the percentage of projecting axons that express the designer receptor. It would have been reassuring to see some electrophysiological proof of the success and extent of the inhibition. Inhibition of the MCN and especially be confirmed electrophysiologically, at least for some examples.

We have carried out additional experiments to provide (i) an estimate of the population of neurons being manipulated and (ii) the physiological effect of the DREADDs. For (i) we have injected retrograde tracer into vlPAG and DREADD into MCN, and counted the proportion of double labelled neurons in MCN. And for (ii) we have recorded from DREADD transfected neurons in cerebellar slices before and during application of CNO.

Experiment (i) is now included in Supplementary Figure 10. The anatomical results from 4 rats show that double labelled neurons in MCN represent, on average 70% and 28% when expressed relative to the PAG and DREADD single labelled populations, respectively.

Experiment (ii) An inhibitory DREADD virus (AAV-hSyn-hM4D(Gi)) was injected bilaterally into the MCN under general anaesthesia. Approximately 4 weeks later we performed current clamp recordings from MCN neurons, in cerebellar slices, to investigate how CNO altered spontaneous firing rate. CNO caused an increase in the spontaneous firing rate in MCN neurons (n = 3, example neuron shown in Author response image 1). In a neuron located in the interpositus, where there was no DREADD expression, we did not see this increase (Author response image 1).

**Author response image 1. sa2fig1:** 

These results may seem surprising, as hM4Di-DREADDs might be expected to cause decreases in firing rate by hyperpolarizing the membrane potential via G-protein inward rectifying potassium channels. However, Locke et al., (2018; see their supplementary figure S6) demonstrate three different responses in lateral cerebellar nuclei following bath applied CNO: an increased firing rate, a decreased firing rate and no change. It is proposed that neurons that have an increase in firing rate, similar to our results, may be caused by disinhibition from local circuitry (see also 1.8).Nevertheless, our brain slice experiments provide evidence that the expression of hM4DiDREADDs in MCN cells has a physiological effect. Additional evidence that the expression of hM4Di-DREADDs is functional is provided by the behavioural deficits we observed when animals were tested on the balance beam following systemic CNO administration.

3.3. The authors don't seem to have performed necessary control experiments to exclude potential effects of CNO on endogenous clozapine binding sites in the brain when using systemic CNO injections.

We thank the reviewer for highlighting our oversight of not including the additional control for any systemic effects of CNO itself. These results are now included in Supplementary Figure 9F which shows control animal (virus without DREADD) beam performance before (baseline) and after systemic CNO injection (blue graph, control). There was no significant difference, consistent with there being no detectable off target effects of the CNO.

3.4. The authors state that MCN inhibition alters the temporal precision of vPAG offset activity. Timing as a contribution of the cerebellum to fear learning is mentioned several times. However, there is no actual data supporting a change in timing or temporal precision. The authors don't analyze the temporal precision of the offset response, only the area under the curve and peak amplitude. The claim that the cerebellum control aspects of timing in this task is not supported by results. The timing of the off response, which is specifically mentioned, needs to be analyzed to support or refute this claim.

Additional quantitative analysis is now included in a new Figure 4 which shows that there is a statistically significant difference between control and muscimol treated animals in both the number and timing of peaks in vPAG activity at CS offset, consistent with an effect on temporal precision of vPAG encoding. See point 2.8 for further details.

3.5. In the last paragraph of the introduction the authors claim that the rate of extinction is affected by MCN inhibition. There are no results to support this. In the same paragraph the authors state that ultrasonic vocalizations (USVs) and rearing behavior during fear acquisition are influenced by MCN inputs to vPAC. However, all results for USV and rearing counts are shown for the extinction phase.

We apologise for the error in wording which has been corrected in light of our increased sample size of behavioural data.

3.6. Experimental findings with an n = 1, as the author's experiment with one rat exposed to trace fear conditioning, are not acceptable. The trace fear conditioning experiment should be repeated with at least two more animals (although a higher number would likely be needed).

We have increased the sample size to 12 units from 3 animals. See 1. 4 for further details.

3.7. Supplementary Figure 1 A is supposed to show electrode tracks for rats implanted with two drives for electrodes. The left cerebellar illustration shows what is declared as canular tracks in the other illustration. Was that by mistake or where cannulas inserted here?

We thank the reviewer for pointing out our mistake in panel position in this figure. We have amended Supplementary Figure 1 and in the legend, we have also make clear which are cannular tracks and which are electrode tracks. See also 1.9 and 2.11.

3.8. The manuscript overstates several findings. For example, the presented findings do not imply that the cerebellum 'controls' any of these behaviors or activity patterns but rather that it is part of a larger circuit that does.

We have amended this, and other statements as suggested.

3.9. Please show some example raw data of your electrophysiological recordings.

We have included example raw tetrode data in Supplementary Figure 2.

[Editors’ note: what follows is the authors’ response to the second round of review.]

Essential revisions:Please see below the specific recommendations by the reviewers. In addition to the points of clarity as requested, please pay particular attention to:1) Cases where animal numbers need to be provided.

Animal numbers have now been included.

2) References as requested.

The additional references have now been included.

3) Details regarding methodology.

More details added to the Methods.

4) Please ensure your manuscript complies with the eLife policies for statistical reporting: https://reviewer.elifesciences.org/author-guide/full "Report exact p-values wherever possible alongside the summary statistics (and degrees of freedom) and 95% confidence intervals. These should be reported for all key questions and not only when the p-value is less than 0.05."Please also include a key resource table.

We have updated all stats to comply with *eLife* policies for statistical reporting

Resource table added.

Reviewer #3:The authors have adequately addressed all my concerns.One small final request would be that the authors mention the results from their test of DREADD activation and the related findings by Locke et al. 2018 in the manuscript. This could just be in writing without adding a figure.

New section added to the Methods (page 27, line 1-17).